# Factored Adaptation for Non-stationary Reinforcement Learning

**Fan Feng**[1], **Biwei Huang**[2], **Kun Zhang**[2,3], **Sara Magliacane**[4,5]
[1]City University of Hong Kong [2]Carnegie Mellon University
[3]Mohamed bin Zayed University of Artificial Intelligence
[4]University of Amsterdam [5]MIT-IBM Watson AI Lab
`{ffeng1017,sara.magliacane}@gmail.com, biweih@andrew.cmu.edu, kunz1@cmu.edu`

## Abstract

Dealing with non-stationarity in environments (e.g., in the transition dynamics) and objectives (e.g., in the reward functions) is a challenging problem that is crucial in real-world applications of reinforcement learning (RL). While most current approaches model the changes as a single shared embedding vector, we leverage insights from the recent causality literature to model non-stationarity in terms of individual latent change factors, and causal graphs across different environments. In particular, we propose Factored Adaptation for Non-Stationary RL (FANS-RL), a factored adaption approach that learns jointly both the causal structure in terms of a factored MDP, and a factored representation of the individual time-varying change factors. We prove that under standard assumptions, we can completely recover the causal graph representing the factored transition and reward function, as well as a partial structure between the individual change factors and the state components. Through our general framework, we can consider general non-stationary scenarios with different function types and changing frequency, including changes across episodes and within episodes. Experimental results demonstrate that FANS-RL outperforms existing approaches in terms of return, compactness of the latent state representation, and robustness to varying degrees of non-stationarity.

## 1 Introduction

Learning a stable policy under non-stationary environments is a long-standing challenge in Reinforcement learning (RL) [1, 2, 3]. While most RL approaches assume stationarity, in many real-world applications of RL there can be changes in the dynamics or the reward function, both *across* different episodes and *within* each episode. Recently, several works adapted Meta-RL methods to learn sequences of non-stationary tasks [4, 5]. However, the continuous MAML [6] adaptation for non-stationary RL [4] does not explicitly model temporal changing components, while TRIO [5] needs to meta-train the model on a set of non-stationary tasks. LILAC [7] and ZeUS [8] leverage latent variable models to directly model the change factors in the environment in a shared embedding space. In particular, they consider families of MDPs indexed by a single latent parameter. In this paper, we argue that disentangling the changes as separate latent parameters and modeling the process with a factored representation improves the efficiency of adapting to non-stationarity.

In particular, we leverage insights from the causality literature [9, 10] that model non-stationarity in terms of individual latent change factors and causal graphs across different environments. We propose Factored Adaptation for Non-Stationary RL (FANS-RL), a factored adaptation framework that jointly learns the causal structure of the MDP and a factored representation of the individual change factors, allowing for changes at discrete timepoints and continuously varying environments. While we provide a specific architecture (FN-VAE), the theoretical framework of FANS-RL can be implemented with different architectures and combined with various RL algorithms. We formalize our setting as a

36th Conference on Neural Information Processing Systems (NeurIPS 2022).

*Factored Non-stationary MDP* (FN-MDP), which combines a Factored-MDP [11, 12, 13] with latent change factors that evolve in time following a Markov process.

We build upon the AdaRL framework [14], a recently proposed fast adaptation approach. AdaRL learns a factored representation that explicitly models changes (i.e., domain-specific components) in observation, dynamics and reward functions across a set of source domains. An optimal policy learnt on the source domains can then be adapted to a new target domain simply by identifying a low-dimensional change factor, without any additional finetuning. FANS-RL extends AdaRL from the stationary case with constant change factors to a general non-stationary framework. Specifically, FANS-RL learns the low-dimensional and time-evolving representations $\boldsymbol{\theta}_t^s$ and $\boldsymbol{\theta}_t^r$ that fully capture the non-stationarity of dynamics and rewards, allowing for continuous and discrete changing functions, both *within-episode* and *across-episode*. Our main contributions can be summarized as:

- We formalize FN-MDPs, a unified factored framework that can handle many non-stationary settings, including discrete and continuous changes, both within and across episodes. We prove that, under standard assumptions, the causal graph of the transition and reward function is identifiable, while we can recover a partial structure for the change factors.
- We introduce Factored Adaptation for Non-Stationary RL (FANS-RL), a general non-stationary RL approach that interleaves model estimation of an FN-MDP and policy optimization. We also describe FN-VAE, an example architecture for learning FN-MDPs.
- We evaluate FANS-RL on simulated benchmarks for continuous control and robotic manipulation tasks and show it outperforms the state of the art on the return, compactness of the latent space representation and robustness to varying degrees of non-stationarity.

## 2    Factored Non-stationary MDPs

To model different types of non-stationarity in a unified and factored way, we propose Factored Non-stationary Markov Decision Processes (FN-MDPs). FN-MDPs are an augmented form of a factored MDPs [11, 12, 13] with latent change factors that evolve over time following a Markov process. Since the change factors are latent, FN-MDPs are partially observed. We define them as:

**Definition 1.** *A Factored Non-stationary Markov Decision Process (FN-MDP) is a tuple $(\mathcal{S}, \mathcal{A}, \Theta^s, \Theta^r, \gamma, \mathcal{G}, \mathbb{P}_s, \mathcal{R}, , \mathbb{P}_{\theta^r}, \mathbb{P}_{\theta^s})$, where $\mathcal{S}$ is the state space, $\mathcal{A}$ the action space, $\Theta^s$ the space of the change factors for the dynamics, $\Theta^r$ the space of the reward change factors and $\gamma$ the discount factor. We assume $\mathcal{G}$ is a Dynamic Bayesian Network over $\{s_{1,t}, ..., s_{d,t}, a_{1,t}, ..., a_{m,t}, r_t, \theta_{1,t}^s, ..., \theta_{p,t}^s, \theta_{1,t}^r, ..., \theta_{q,t}^r\}$, where d, m, p, and q are the dimensions of states, action, change factors on dynamics and reward, respectively. We define the factored state transition distribution $\mathbb{P}_s$ as:*

$$\mathbb{P}_s(\boldsymbol{s}_t|\boldsymbol{s}_{t-1}, \boldsymbol{a}_{t-1}, \boldsymbol{\theta}_t^s) = \prod_{i=1}^{d} \mathbb{P}_s(s_{i,t}|pa(s_{i,t}))$$

*where $pa(s_{i,t})$ denotes the causal parents of $s_{i,t}$ in $\mathcal{G}$, which are a subset of the dimensions of $\boldsymbol{s}_{t-1}$, $\boldsymbol{a}_{t-1}$ and $\boldsymbol{\theta}_t^s$. Note that the action $\boldsymbol{a}_{t-1}$ is a vector of m dimensions in our setting. We assume a given initial state distribution $\mathbb{P}_s(\boldsymbol{s}_0)$. Similarly, we define the reward function $\mathcal{R}$ as a function of the parents of $r_t$ in $\mathcal{G}$, i.e., $\mathcal{R}(\mathbf{s}_t, \mathbf{a}_t, \boldsymbol{\theta}_t^r) = \mathcal{R}(pa(r_t))$, where $pa(r_t)$ are a subset of dimensions of $\mathbf{s}_t, \mathbf{a}_t,$ and $\boldsymbol{\theta}_t^r$. We define the factored latent change factors transition distributions $\mathbb{P}_{\theta^s}$ and $\mathbb{P}_{\theta^r}$ as:*

$$\mathbb{P}_{\theta^s}(\boldsymbol{\theta}_t^s|\boldsymbol{\theta}_{t-1}^s) = \prod_{j=1}^{p} \mathbb{P}_{\theta^s}(\theta_{j,t}^s|pa(\theta_{j,t}^s)), \qquad \mathbb{P}_{\theta^r}(\boldsymbol{\theta}_t^r|\boldsymbol{\theta}_{t-1}^r) = \prod_{k=1}^{q} \mathbb{P}_{\theta^r}(\theta_{k,t}^r|pa(\theta_{k,t}^r))$$

*where $pa(\theta_{j,t}^s)$ are a subset of the dimensions of $\boldsymbol{\theta}_{t-1}^s$, while $pa(\theta_{k,t}^r)$ are a subset of dimensions of $\boldsymbol{\theta}_{t-1}^r$. We assume the initial distributions $\mathbb{P}_{\theta^s}(\boldsymbol{\theta}_0^s)$ and $\mathbb{P}_{\theta^r}(\boldsymbol{\theta}_0^r)$ are given.*

We show an example DBN representing the graph $\mathcal{G}$ of an FN-MDP in Fig. 1(a). Since we are interested in learning the graphical structure of the FN-MDP, as well as identifying the values of the latent change factors from data, we describe a generative process of an FN-MDP environment. In particular, we assume that the graph $\mathcal{G}$ is time-invariant throughout the non-stationarity, and there are no unobserved confounders and instantaneous causal effects in the system. We will learn a set of binary masks $\boldsymbol{c}^{\cdot\rightarrow\cdot}$ and $\boldsymbol{C}^{\cdot\rightarrow\cdot}$ that are the indicators for edges in $\mathcal{G}$.

**Generative environment model.** We adapt the generative model in AdaRL [14] across $k$ different domains to a time-varying setting on a single domain. We assume the generative process of the environment at timestep $t$ in terms of the transition function for each dimension $i = 1, ..., d$ of $\mathbf{s}_t$ is:

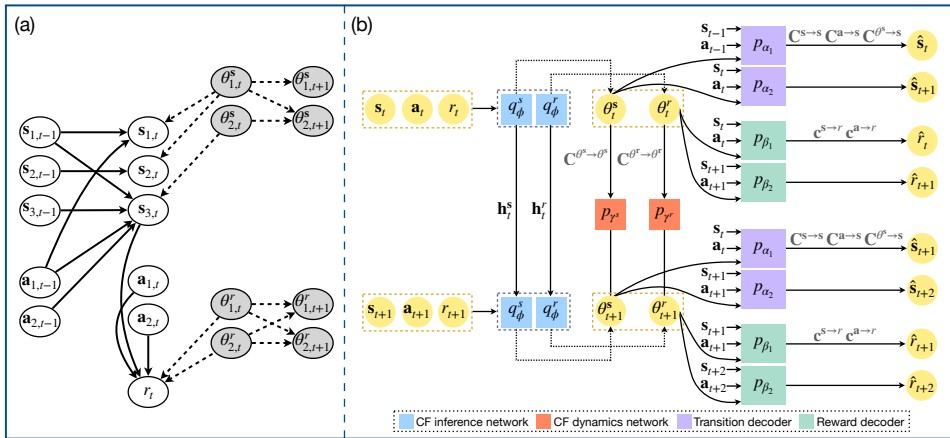

Figure 1: (a). A graphical representation of an FN-MDP. For readability, we only illustrate a subsection of dimensions of states, actions, and latent change factors. The shaded variables are unobserved; (b). The architecture of FN-VAE, which learns the generative model, explained in Sec. 3.

$$s_{i,t} = f_i\big(\boldsymbol{c}_i^{\boldsymbol{s} \to \boldsymbol{s}} \odot \boldsymbol{s}_{t-1},\, \boldsymbol{c}_i^{\boldsymbol{a} \to \boldsymbol{s}} \odot \boldsymbol{a}_{t-1},\, \boldsymbol{c}_i^{\boldsymbol{\theta}_t \to \boldsymbol{s}} \odot \boldsymbol{\theta}_t^{\boldsymbol{s}},\, \epsilon_{i,t}^s\big) \tag{1}$$

where $\odot$ is the element-wise product, $f_i$ are non-linear functions and $\epsilon_{i,t}^s$ is an i.i.d. random noise. The binary mask $\boldsymbol{c}_i^{\boldsymbol{s} \to \boldsymbol{s}} \in \{0,1\}^d$ represents which state components $s_{j,t-1}$ are used in the transition function of $s_{i,t}$. Similarly, $\boldsymbol{c}_i^{\boldsymbol{a} \to \boldsymbol{s}} \in \{0,1\}^m$ indicates whether the action directly affects $s_{i,t}$. The change factor $\boldsymbol{\theta}_t^{\boldsymbol{s}} \in \mathbb{R}^p$ encodes any change in the dynamics. The binary mask $\boldsymbol{c}_i^{\boldsymbol{\theta}_t \to \boldsymbol{s}} \in \{0,1\}^p$ represents which of the components of $\boldsymbol{\theta}_t^{\boldsymbol{s}}$ influence $s_{i,t}$. We model the reward function as:

$$r_t = h\big(\boldsymbol{c}^{\boldsymbol{s} \to r} \odot \boldsymbol{s}_t,\, \boldsymbol{c}^{\boldsymbol{a} \to r} \odot \boldsymbol{a}_t,\, \boldsymbol{\theta}_t^r,\, \epsilon_t^r\big) \tag{2}$$

where $\boldsymbol{c}^{\boldsymbol{s} \to r} \in \{0,1\}^d$, $\boldsymbol{c}^{\boldsymbol{a} \to s} \in \{0,1\}^m$, and $\epsilon_t^r$ is an i.i.d. random noise. The change factor $\boldsymbol{\theta}_t^r \in \mathbb{R}^q$ encodes any change in the reward function. The binary masks $\boldsymbol{c}^{\to \cdot}$ can be seen as indicators of edges in the DBN $\mathcal{G}$. In AdaRL, all change factors are assumed to be constant in each domain. Since in this paper we allow the change parameters to evolve in time, we introduce two additional equations:

$$\begin{aligned} \theta_{j,t}^s &= g^s\big(\boldsymbol{c}_j^{\boldsymbol{\theta}^s \to \boldsymbol{\theta}^s} \odot \boldsymbol{\theta}_{t-1}^{\boldsymbol{s}},\, \epsilon_t^{\theta^s}\big) \\ \theta_{k,t}^r &= g^r\big(\boldsymbol{c}_k^{\boldsymbol{\theta}^r \to \boldsymbol{\theta}^r} \odot \boldsymbol{\theta}_{t-1}^r,\, \epsilon_t^{\theta^r}\big) \end{aligned} \tag{3}$$

for $i = 1, ..., d$, $j = 1, ..., p$, $k = 1, ..., q$, and $g^s$, and $g^r$ are non-linear functions. We assume the binary masks $\boldsymbol{c}^{\to \cdot}$ are stationary across timesteps and so are the $\epsilon_{i,t}^s, \epsilon_t^r, \epsilon_t^{\theta^s}$ and $\epsilon_t^{\theta^r}$, the i.i.d. random noises. Although $\boldsymbol{c}^{\to \cdot}$ and $\epsilon$ are stationary, we model the changes in the functions and some changes in the graph structure through $\boldsymbol{\theta}$. For example a certain value of $\boldsymbol{\theta}_t^r$ can switch off the contribution of some of the state or action dimensions in the reward function, or in other words nullify the effect of some edges in $\mathcal{G}$. Similarly the contribution of the noise distribution to each function can be modulated via the change factors. On the other hand, this setup does not allow adding edges that are not captured by the binary masks $\boldsymbol{c}^{\to \cdot}$. We group the binary masks in the matrices $\boldsymbol{C}^{\boldsymbol{s} \to \boldsymbol{s}} := [\boldsymbol{c}_i^{\boldsymbol{s} \to \boldsymbol{s}}]_{i=1}^d$, $\boldsymbol{C}^{\boldsymbol{\theta}^s \to \boldsymbol{s}} := [\boldsymbol{c}_i^{\boldsymbol{\theta}^s \to \boldsymbol{s}}]_{i=1}^d$, and $\boldsymbol{C}^{\boldsymbol{a} \to \boldsymbol{s}} := [\boldsymbol{c}_i^{\boldsymbol{a} \to \boldsymbol{s}}]_{i=1}^d$. Similarly, we also group the binary vectors in the dynamics of the latent change factors in matrices $\boldsymbol{C}^{\boldsymbol{\theta}^s \to \boldsymbol{\theta}^s} := [\boldsymbol{c}_j^{\boldsymbol{\theta}^s \to \boldsymbol{\theta}^s}]_{j=1}^p$ and $\boldsymbol{C}^{\boldsymbol{\theta}^r \to \boldsymbol{\theta}^r} := [\boldsymbol{c}_k^{\boldsymbol{\theta}^r \to \boldsymbol{\theta}^r}]_{k=1}^q$. Since latent change factors $\boldsymbol{\theta}^s$ and $\boldsymbol{\theta}^r$ follow a Markov process based on $g^s$ and $g^r$, we can consider different types of changes by varying the form of $g^s$ and $g^r$, generalizing the approaches in literature. We can also model concurrent changes in dynamics and reward, including different types of changes, e.g. a continuous $g^s$ and a piecewise-constant $g^r$.

**Compact representations.** Huang et al. [14] show that the only dimensions of the state and change factors useful for policy learning are those that eventually affect the reward. These dimensions are called *compact representations* and are defined as the dimensions of the state and change factors with

a path (i.e. a sequence of edges $\rightarrow$) to the present or future reward $r_{t+\tau}$ for $\tau \geq 0$ in the DBN $\mathcal{G}$:

$$s_{i,t} \in \boldsymbol{s}^{min} \iff s_{i,t} \rightarrow \ldots \rightarrow r_{t+\tau} \text{ for } \tau \geq 0, \text{ and } \theta_i \in \boldsymbol{\theta}^{min} \iff \theta_i \rightarrow \ldots \rightarrow r_{t+\tau} \text{ for } \tau \geq 0$$

**Continuous changes.** If $g^s$ and $g^r$ are continuous, then they can model smooth changes in the environment, including across episodes. While the functions in Eq. 1-3 allow us to model *within-episode* changes, i.e. changes that can happen only before $t = H$ where $H$ is the horizon, we also want to model *across-episode* changes. We use a separate time index $\tilde{t}$ that models the agent's lifetime. Initially $\tilde{t} = t$ for $t \leq H$, but while we reset $t = 0$ afterwards, $\tilde{t}$ continues to grow indefinitely.

**Discrete changes.** We assume discrete changes happen at specific timesteps and can be represented with a piecewise constant function. In particular we denote change timesteps as $\tilde{\boldsymbol{t}} = (\tilde{t}_1, \ldots, \tilde{t}_M)$ where $\tilde{t}_i$ describes a specific timepoint in the agent's lifetime time index $\tilde{t}$. This allows us to model *within-episode changes*. In this case, we assume that the change happens always at the same steps $(t_1, \ldots, t_m)$ in each episode, i.e., we assume that $\tilde{\boldsymbol{t}} = (t_1, \ldots, t_m, H + t_1, \ldots, H + t_m, 2H + t_1, \ldots, 2H + t_m, \ldots)$, where $H$ is the horizon. We can also model *across-episode changes*, when we assume the change points only occur at the end of each episode, i.e., $\tilde{\boldsymbol{t}} = (H, 2H, 3H, \ldots)$.

We extend the result on identifiability of factored MDPs in AdaRL [14] to non-stationary environments. We first assume that we observe the change factors and show we can identify the true causal graph $\mathcal{G}$:

**Proposition 1** (Full identifiability with observed change factors)**.** *Suppose the generative process follows Eq. 1-3 and all change factors $\boldsymbol{\theta}_t^s$ and $\boldsymbol{\theta}_t^r$ are observed, i.e., Eq. 1-3 is an MDP. Under the Markov and faithfulness assumptions, i.e. conditional independences correspond exactly to d-separations, all binary masks $\boldsymbol{C}^{\cdot \rightarrow \cdot}$ are identifiable, i.e., we can fully recover the causal graph $\mathcal{G}$.*

We provide all proofs and a detailed explanation in Appendix B. If we do not observe the change factors $\boldsymbol{\theta}_t^s$ and $\boldsymbol{\theta}_t^r$, we cannot identify their dimensions, and we cannot fully recover the causal graph $\mathcal{G}$. On the other hand, we can still identify the partial causal graph over the state variables $\boldsymbol{s}_t$, reward variable $r_t$, and action variable $\boldsymbol{a}_t$. We can also identify which dimensions in $s_{i,t}$ have changes, i.e., we can identify $\boldsymbol{C}^{\boldsymbol{\theta}^s \rightarrow \boldsymbol{s}}$. We formalize this idea in the following (proof in Appendix B):

**Proposition 2** (Partial Identifiability with latent change factors)**.** *Suppose the generative process follows Eq. 1-3 and the change factors $\boldsymbol{\theta}_t^s$ and $\boldsymbol{\theta}_t^r$ are unobserved. Under the Markov and faithfulness assumptions, the binary masks $\boldsymbol{C}^{\boldsymbol{s} \rightarrow \boldsymbol{s}}, \boldsymbol{C}^{\boldsymbol{a} \rightarrow \boldsymbol{s}}, \boldsymbol{c}^{\boldsymbol{s} \rightarrow r}$ and $\boldsymbol{c}^{\boldsymbol{a} \rightarrow r}$ are identifiable. Moreover, we can identify which state dimensions are affected by $\boldsymbol{\theta}_t^s$ and whether the reward function changes.*

This means that even in the most general case, we can learn most of the true causal graph $G$ in an FN-MDP, with the exception of the transition structure of the latent change factors. In the following, we show a variational autoencoder setup to learn the generative process in FN-MDPs.

## 3   Learning the Generative Process in FN-MDPs

There are many possible architectures to learn FN-MDPs through Eq. 1-3. We propose *FN-VAE*, a variational autoencoder architecture described in Fig. 1(b). In FN-VAE, we jointly learn the structural relationships, state transition function, reward function, and transition function of the latent change factors, as described in detail in Appendix Alg. A1. An FN-VAE has four types of components: *change factor (CF) inference networks* that reconstruct the latent change factors, *change factor (CF) dynamics networks* that model their dynamics with an LSTM [15], *transition decoders* that reconstruct the state dynamics at the time $t$ and predict one step further at $t + 1$, and *reward decoders* that reconstruct the reward at $t$ and predict the future reward at $t + 1$. We now describe them in detail.

**CF inference networks (blue boxes in Fig. 1(b)).** The two inference models for latent change factors $q_{\phi^s}(\boldsymbol{\theta}_t^s \mid \boldsymbol{s}_t, \boldsymbol{a}_t)$ and $q_{\phi^r}(\boldsymbol{\theta}_t^r \mid \boldsymbol{s}_t, \boldsymbol{a}_t, r_t)$ are parameterised by $\phi^s$ and $\phi^r$, respectively. To model the time-dependence of $\boldsymbol{\theta}_t^s$ and $\boldsymbol{\theta}_t^r$, we use LSTMs [15] as inference networks. At timestep $t$, the dynamics change factor LSTM infers $q_{\phi^s}(\boldsymbol{\theta}_t^s \mid \boldsymbol{s}_t, \boldsymbol{a}_t, r_t, \boldsymbol{h}_{t-1}^s)$, where $\boldsymbol{h}_{t-1}^s \in \mathbb{R}^L$ is the hidden state in the LSTM. Thus we can obtain $\mu_{\phi^s}(\boldsymbol{\tau}_{0:t})$ and $\sigma_{\phi^s}^2(\boldsymbol{\tau}_{0:t})$ using $q_{\phi^s}$, and sample the latent changing factor $\boldsymbol{\theta}_t^s \sim \mathcal{N}(\mu_{\phi^s}(\boldsymbol{\tau}_{0:t}), \sigma_{\phi^s}^2(\boldsymbol{\tau}_{0:t}))$, where $\boldsymbol{\tau}_{0:t} = (\boldsymbol{s}_0, \boldsymbol{a}_0, r_0, \boldsymbol{s}_1, \boldsymbol{a}_1, r_2, \ldots, \boldsymbol{s}_t, \boldsymbol{a}_t, r_t)$. Similarly, the reward change factor LSTM infers $q_{\phi^r}(\boldsymbol{\theta}_t^r \mid \boldsymbol{s}_t, \boldsymbol{a}_t, r_t, \boldsymbol{h}_{t-1}^r)$, where $\boldsymbol{h}_{t-1}^r \in \mathbb{R}^L$ is the hidden state, such that we can sample $\boldsymbol{\theta}_t^r \sim \mathcal{N}(\mu_{\phi^r}(\boldsymbol{\tau}_{0:t}), \sigma_{\phi^r}^2(\boldsymbol{\tau}_{0:t}))$.

**CF dynamics network (orange boxes in Fig. 1(b)).** We model the dynamics of latent change factors with $p_{\gamma^s}(\boldsymbol{\theta}_{t+1}^s \mid \boldsymbol{\theta}_t^s, \boldsymbol{C}^{\boldsymbol{\theta}^s \rightarrow \boldsymbol{\theta}^s})$ and $p_{\gamma^r}(\boldsymbol{\theta}_{t+1}^r \mid \boldsymbol{\theta}_t^r, \boldsymbol{C}^{\boldsymbol{\theta}^r \rightarrow \boldsymbol{\theta}^r})$. To ensure the Markovianity of $\boldsymbol{\theta}_t^s$ and

$\boldsymbol{\theta}_t^r$, we define a loss $\mathcal{L}_{\mathrm{KL}}$ that helps minimize the KL-divergence between $q_\phi$ and $p_\gamma$.

$$\mathcal{L}_{\mathrm{KL}} = \sum_{t=2}^{T} \mathrm{KL}\big(q_{\phi^s}\big(\boldsymbol{\theta}_t^s \mid \boldsymbol{s}_t, \boldsymbol{a}_t, r_t, \boldsymbol{h}_{t-1}^s\big)\|p_{\gamma^s}(\boldsymbol{\theta}_t^s|\boldsymbol{\theta}_{t-1}^s; \boldsymbol{C}^{\boldsymbol{\theta}^s \rightarrow \boldsymbol{\theta}^s})\big) \tag{4}$$
$$+ \mathrm{KL}\big(q_{\phi^r}(\boldsymbol{\theta}_t^r \mid \boldsymbol{s}_t, \boldsymbol{a}_t, r_t, \boldsymbol{h}_{t-1}^r))\|p_{\gamma^r}(\boldsymbol{\theta}_t^r|\boldsymbol{\theta}_{t-1}^r; \boldsymbol{C}^{\boldsymbol{\theta}^r \rightarrow \boldsymbol{\theta}^r})\big)$$

If we assume that the change between $\boldsymbol{\theta}_t^s$ and $\boldsymbol{\theta}_{t+1}^s$, and similarly $\boldsymbol{\theta}_t^r$, is smooth, we can add a smoothness loss $\mathcal{L}_{\mathrm{smooth}}$. We provide a smooth loss for discrete changes in Appendix D.2.

$$\mathcal{L}_{\mathrm{smooth}} = \sum_{t=2}^{T} \big(||\boldsymbol{\theta}_t^s - \boldsymbol{\theta}_{t-1}^s||_1 + ||\boldsymbol{\theta}_t^r - \boldsymbol{\theta}_{t-1}^r||_1\big) \tag{5}$$

**Transition decoders (purple boxes in Fig. 1(b)).** We learn an approximation of the transition dynamics in Eq. 1 by learning a reconstruction, parameterized by $\alpha_1$, and a prediction encoder, parametrized by $\alpha_2$. To simplify the formulas, we define $\boldsymbol{C}^{\cdot \rightarrow s} := (\boldsymbol{C}^{s \rightarrow s}, \boldsymbol{C}^{a \rightarrow s}, \boldsymbol{C}^{\boldsymbol{\theta}^s \rightarrow s})$. At timestep $t$, the reconstruction encoder $p_{\alpha_1}(\boldsymbol{s}_t \mid \boldsymbol{s}_{t-1}, \boldsymbol{a}_{t-1}, \boldsymbol{\theta}_t^s; \boldsymbol{C}^{\cdot \rightarrow s})$ reconstructs the state from current state $\boldsymbol{s}_t$ with sampled $\boldsymbol{\theta}_t^s$. The one-step prediction encoder $p_{\alpha_2}(\boldsymbol{s}_{t+1} \mid \boldsymbol{s}_t, \boldsymbol{a}_t, \boldsymbol{\theta}_t^s)$ instead tries to approximate the next state $\boldsymbol{s}_{t+1}$. We do not use the prediction loss, when the one-step prediction is not smooth. In particular, we do not use it for the last time-step in episode $i$ if there is a change happening at the first step in episode $(i + 1)$, since the states in new episodes will be randomly initiated. We also do not use it in the case of discrete changes at the timesteps $(\tilde{t}_1 - 1, \ldots, \tilde{t}_M - 1)$. The loss functions are:

$$\mathcal{L}_{\mathrm{rec\text{-}dyn}} = \sum_{t=1}^{T-2} \mathbb{E}_{\theta_t^s \sim q_\phi} \log p_{\alpha_1}(\boldsymbol{s}_t|\boldsymbol{s}_{t-1}, \boldsymbol{a}_{t-1}, \boldsymbol{\theta}_t^s; \boldsymbol{C}^{\cdot \rightarrow s})$$
$$\mathcal{L}_{\mathrm{pred\text{-}dyn}} = \sum_{t=1}^{T-2} \mathbb{E}_{\theta_t^s \sim q_\phi} \log p_{\alpha_2}(\boldsymbol{s}_{t+1}|\boldsymbol{s}_t, \boldsymbol{a}_t, \boldsymbol{\theta}_t^s) \tag{6}$$

**Reward decoders (green boxes in Fig. 1(b)).** Similarly, we use a reconstruction encoder $p_{\beta_1}(r_t \mid \boldsymbol{s}_t, \boldsymbol{a}_t, \boldsymbol{\theta}_t^r, \boldsymbol{c}^{s \rightarrow r}, \boldsymbol{c}^{a \rightarrow r})$, parameterized by $\beta_1$, and a one-step prediction encoder $p_{\beta_2}(r_{t+1} \mid \boldsymbol{s}_{t+1}, \boldsymbol{a}_{t+1}, \boldsymbol{\theta}_t^r)$, parametrized by $\beta_2$, to approximate the reward function. Similarly to transition decoders, we do not use the one-step prediction loss, if it is not smooth. The losses are:

$$\mathcal{L}_{\mathrm{rec\text{-}rw}} = \sum_{t=1}^{T-2} \mathbb{E}_{\theta_t^r \sim q_\phi} \log p_{\beta_1}(r_t|\boldsymbol{s}_t, \boldsymbol{a}_t, \boldsymbol{\theta}_t^r; \boldsymbol{c}^{s \rightarrow r}, \boldsymbol{c}^{a \rightarrow r})$$
$$\mathcal{L}_{\mathrm{pred\text{-}rw}} = \sum_{t=1}^{T-2} \mathbb{E}_{\theta_t^r \sim q_\phi} \log p_{\beta_2}(r_{t+1}|\boldsymbol{s}_{t+1}, \boldsymbol{a}_{t+1}, \boldsymbol{\theta}_t^r) \tag{7}$$

**Sparsity loss.** We encourage sparsity in the binary masks $\boldsymbol{C}^{\cdot \rightarrow \cdot}$ to improve identifiability, by using following loss with adjustable hyperparameters $(w_1, \ldots, w_7)$, which we learn through grid search.

$$\mathcal{L}_{\mathrm{sparse}} = w_1\|\boldsymbol{C}^{s \rightarrow s}\|_1 + w_2\|\boldsymbol{C}^{a \rightarrow s}\|_1 + w_3\|\boldsymbol{C}^{\boldsymbol{\theta}^s \rightarrow s}\|_1 + w_4\|\boldsymbol{c}^{s \rightarrow r}\|_1 + w_5\|\boldsymbol{c}^{a \rightarrow r}\|_1$$
$$+ w_6\|\boldsymbol{C}^{\boldsymbol{\theta}^s \rightarrow \boldsymbol{\theta}^s}\|_1 + w_7\|\boldsymbol{C}^{\boldsymbol{\theta}^r \rightarrow \boldsymbol{\theta}^r}\|_1 \tag{8}$$

The total loss is $\mathcal{L}_{\mathrm{vae}} = k_1(\mathcal{L}_{\mathrm{rec\text{-}dyn}} + \mathcal{L}_{\mathrm{rec\text{-}rw}}) + k_2(\mathcal{L}_{\mathrm{pred\text{-}dyn}} + \mathcal{L}_{\mathrm{pred\text{-}rw}}) - k_3\mathcal{L}_{\mathrm{KL}} - k_4\mathcal{L}_{\mathrm{sparse}} - k_5\mathcal{L}_{\mathrm{smooth}}$, where $(k_1, \ldots, k_5)$: hyper-parameters, which we learn with an automatic weighting method [16].

**Learning from raw pixels.** Our framework can be easily extended to image inputs by adding an encoder $\phi^o$ to learn the latent state variables from pixels, similar to other works [17, 18, 14]. In this case, our identifiability results do not hold anymore, since we cannot guarantee that we identify the true causal variables. We describe this component in Appendix D.3.

## 4 FANS-RL: Online Model Estimation and Policy Optimization

We propose Factored Adaptation for Non-Stationary RL (FANS-RL), a general algorithm that interleaves model estimation and policy optimization, as shown in Alg. 1 and Appendix Fig. A9.

---

**Algorithm 1:** Factored Adaptation for non-stationary RL

---

1: **Init:** Env; VAE parameters: $\phi = (\phi^s, \phi^r)$, $\alpha = (\alpha_1, \alpha_2)$, $\beta = (\beta_1, \beta_2)$, $\gamma$; Binary masks: $\boldsymbol{C}^{\cdot \rightarrow \cdot}$;
    Policy parameters: $\psi$; replay buffer: $\mathcal{D}$; Number of episodes: $N$; Episode horizon: $H$; Initial $\boldsymbol{\theta}$:
    $\boldsymbol{\theta}^s_{\text{old}}$ and $\boldsymbol{\theta}^r_{\text{old}}$; Length of collected trajectory: $k$.
2: **Output:** VAE parameters: $\phi, \alpha, \beta, \gamma$; Policy parameters: $\psi$
3: Collect multiple trajectories of length $k$ : $\boldsymbol{\tau} = \{\boldsymbol{\tau}^1_{0:k}, \boldsymbol{\tau}^2_{0:k}, \ldots\}$ with policy $\pi_\psi$ from Env;
4: Learn FN-VAE from $\boldsymbol{\tau}$, including masks $\boldsymbol{C}^{\cdot \rightarrow \cdot}$ that represent the graph $\mathcal{G}$ (Appendix Alg. A1)
5: Identify the compact representations $\boldsymbol{s}^{min}$ and change factors $\boldsymbol{\theta}^{min}$ based on $\boldsymbol{C}^{\cdot \rightarrow \cdot}$
6: **for** $n = 0, \ldots, N - 1$ **do**
7:     **for** $t = 0, \ldots, H - 1$ **do**
8:        Observe $\boldsymbol{s}_t$ from Env;
9:        **if** $t = 0$ **then**
10:          $\boldsymbol{\theta}^s \leftarrow \boldsymbol{\theta}^s_{\text{old}}$ and $\boldsymbol{\theta}^r \leftarrow \boldsymbol{\theta}^r_{\text{old}}$
11:        **else**
12:          $\boldsymbol{\theta}^s \leftarrow \boldsymbol{\theta}^s_{t-1}$ and $\boldsymbol{\theta}^r \leftarrow \boldsymbol{\theta}^r_{t-1}$
13:        **end if**
14:        **for** j = s, r **do**
15:          Infer mean $\mu_{\gamma^j}(\boldsymbol{\theta}^j)$ and variance $\sigma^2_{\gamma^j}(\boldsymbol{\theta}^j)$ of the change parameter $\boldsymbol{\theta}^j_t$ via $p_{\gamma^j}$
16:          Sample $\boldsymbol{\theta}^j_t \sim \mathcal{N}\left(\mu_{\gamma^j}(\boldsymbol{\theta}^j), \sigma^2_{\gamma^j}(\boldsymbol{\theta}^j)\right)$
17:        **end for**
18:        **if** $t = H - 1$ **then**
19:          $\boldsymbol{\theta}^s_{\text{old}} \leftarrow \boldsymbol{\theta}^s_t$ and $\boldsymbol{\theta}^r_{\text{old}} \leftarrow \boldsymbol{\theta}^s_t$;
20:        **end if**
21:        Generate $\boldsymbol{a}_t \sim \pi_\psi(\boldsymbol{a}_t \mid \boldsymbol{s}^{min}_t, \boldsymbol{\theta}^{min}_t)$ and receive reward $r_{n,t}$ from Env;
22:        Add $(\boldsymbol{s}_t, \boldsymbol{a}_t, r_t, \boldsymbol{\theta}^s_t, \boldsymbol{\theta}^r_t)$ to replay buffer $\mathcal{D}$;
23:        Extract a trajectory with length $k$ from replay buffer $\mathcal{D}$;
24:        Learn FN-VAE (Appendix Alg. A1) with updateG=False (i.e. with fixed masks $\boldsymbol{C}^{\cdot \rightarrow \cdot}$);
25:        Sample a batch of data from replay buffer $\mathcal{D}$ and update policy network parameters $\psi$;
26:     **end for**
27: **end for**

---

After we estimate the initial FN-MDP with the FN-VAE, we can identify compact representations $\boldsymbol{s}^{min}$ and $\boldsymbol{\theta}^{min}$ following AdaRL [14]. In particular, the only dimensions of the state and change factors that are useful for policy learning are those that have a directed path to the reward in the graph $\mathcal{G}$. The online policy $\pi_\psi\left(\boldsymbol{a}_t \mid \boldsymbol{s}^{min}_t, q_\phi(\boldsymbol{\theta}^{min}_t \mid \boldsymbol{\tau}_{0:t})\right)$ can be learned end-to-end, including learning the FN-VAE, as shown in Alg. 1. We use SAC [19] as our policy learning model, so the policy parameters are $\psi = (\pi, Q)$.

**Continuous changes.** In Alg. 1 we describe our framework in case of continuous changes that can span across episodes. We start by collecting a few trajectories $\tau$ and then learn our initial FN-VAE (Lines 3-4). We can use the graphical structure of the initial FN-VAE to identify the compact representations (Line 5). During the online model estimation and policy learning stage, we estimate the latent change factors $\boldsymbol{\theta}^s_t$ and $\boldsymbol{\theta}^r_t$ using the CF dynamics networks $\gamma^s$ and $\gamma^r$ (Lines 8-20). Since in this case, we assume the dynamics of the change factors are smooth across episodes, at time $t = 0$ we will use the last timestep $(H - 1)$ of the previous episode as a prior on the change factors (Line 10). Otherwise, we will estimate the change factors using their values in the previous timestep $t - 1$. We use the estimated latent factors $\boldsymbol{\theta}_t$ and observed state $\boldsymbol{s}_t$ to generate $\boldsymbol{a}_t$ using $\pi_\psi$ and receive a reward $r_t$ (Line 21). We add $(\boldsymbol{s}_t, \boldsymbol{a}_t, r_t, \boldsymbol{\theta}^s_t, \boldsymbol{\theta}^r_t)$ to the replay buffer (Line 22). We now update our estimation of the FN-VAE, but we keep the graph $G$ fixed (Lines 23-24). Finally, we sample a batch of trajectories in the replay buffer and update the policy network $\psi$ (Line 25).

**Discrete changes.** Since we assume discrete changes happen at specific timestep $\tilde{\boldsymbol{t}} = (\tilde{t}_1, \ldots, \tilde{t}_M)$, we can easily modify Alg. 1 for discrete changes, both within-episode and across-episode, by changing Lines 9-20 to only update the change parameters at the timesteps in $\tilde{\boldsymbol{t}}$, as shown in Appendix Alg. A2.

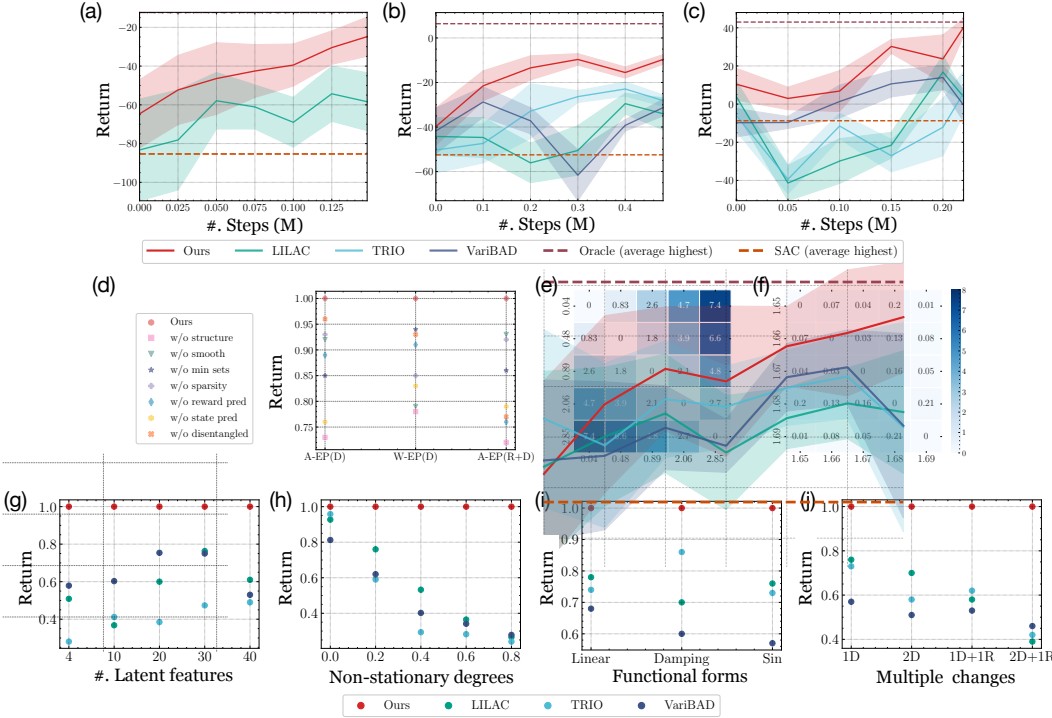

Figure 2: Summary of experimental results. (a)-(c). Average return (smoothed) across 10 runs. We only indicate the average of the highest result of all times for oracle and SAC. The shaded region is $(\mu - \sigma, \mu + \sigma)$, where $\mu$ is the mean and $\sigma$ is the standard deviation. (a) Half-Cheetah-V3 with continuous (sine) changes on $f_w$; (b) Sawyer-Reaching with discrete across-episode changes on $s^g$; and (c) Minitaur with discrete across-episode changes on $m$ and $s_{t,v}$ concurrently. (d) Ablation studies on Half-Cheetah with across & within episode changes on dynamics and across episode changes on both dynamics and rewards. (e)-(f): Pairwise distance on learned $\boldsymbol{\theta}$ between different time steps in Half-Cheetah experiment with across-episode changes on rewards. (g)-(j): Average and normalized final return on 10 runs on Half-Cheetah (g) with within-in episode changes on wind forces using a different number of dimensions in latent representation space; (h) with different non-stationary degrees on across-episode and multi-factor changes; (i) with different functional forms (across episode changes on dynamics); and (j) with different combinations of across-episode changes.

## 5   Evaluation

We evaluate our approach on four well-established benchmarks, including Half-Cheetah-V3 from MuJoCo [20, 21], Sawyer-Reaching and Sawyer-Peg from Sawyer [22, 18], and Minitaur [23]. We modified these tasks to test several non-stationary RL scenarios with continuous and discrete changes. The results suggest that FANS-RL can (1) obtain high rewards, (2) learn meaningful mappings to the change factors with compact latent vectors, and (3) be robust to different non-stationary levels, different functional forms (e.g. piecewise linear, sine, damped sine) and multiple concurrent changes. For space limits, we only highlight a subset of results, and report the full results in Appendix C.4. The implementation will be open-sourced at https://bit.ly/3erKoWm.

**Half-Cheetah-v3 in MuJoCo.** In this task, the agent is moving forward using the joint legs and the objective is to achieve the target velocity $v^g$. We consider both the changes in the dynamics (change of the wind forces $f^w$, change of gravity) and reward functions (change of target velocity $v^g$). We also consider changes on the agent's mechanism, where one random joint is disabled. The reward function is $r_t = -\|v_t^o - v_t^g\|_2 - 0.05\|a_t\|_2$, where $v^o$ and $a_t$ are the agent's velocity and action, respectively, at timestep $t$. The number of time steps in each episode is 50. For dynamics, we change the wind forces $f^w$ in the environment. Moreover, in terms of the reward functions, we change the target velocity $v^g$ to be a time-dependent variable. The change function in the dynamics $f^w$ can be either continuous or discrete, and discrete changes can happen both within and across episodes.

| Methods | ZeUS | Meld | CaDM | Hyper-Dynamics | Ours |
|---|---|---|---|---|---|
| Best Avg. Return | 12.45 | 6.38 | 4.04 | 3.91 | **18.01** |

Table 1: Average highest return on Sawyer-Peg. The number of random trails is 10.

Similarly to LILAC [7], we choose different functions (piecewise linear, sine and damped sine), besides allowing the change at specified intervals, we also allow it to change smoothly. The change in the reward function $v^g$ is not generally stable in the continuous case, so we only consider the discrete and across episode change functions $v^g$ for the reward. We also design a scenario where dynamic and reward functions change concurrently. We report all equations for $g^s$ and $g^r$ in Appendix C.1.

**Sawyer.** We consider two robotic manipulation tasks, Sawyer-Reaching and Sawyer-Peg. We describe the non-stationary settings in Appendix C.2.

In Sawyer-Reaching, the sawyer arm is trained to reach a target position $s^g$. The reward $r_t$ is the difference between the current position $s_t$ and the target position $r_t = -\|s_t - s^g\|_2$. In this task, we cannot directly modify the dynamics in the simulator, so consider a reward-varying scenario, where the target location changes across each episode following a periodic function.

In Sawyer-Peg, the robot arm is trained to insert a peg into a designed target location $s^g$. In this task, following [8], we consider a reward-varying scenario, where the target location changes across each episode following a periodic function. In order to compare with similar approaches, e.g., ZeUS [8], CADM [24], Hyperdynamics [25] and Meld [18], we evaluate our method on raw pixels. Following [8], we consider discrete across-episode changes, where the target location can change in each episode, and is randomly sampled from a small interval.

**Minitaur.** A minitaur robot is a simulated quadruped robot with eight direct-drive actuators. The minitaur is trained to move at the target speed $s_{t,v}$. The reward is $r_t = 0.3 - |0.3 - s_{t,v}| - 0.01 \cdot \|a_t - 2a_{t-1} + a_{t-2}\|_1$. We modify (1) the mass $m$ (dynamics) and (2) target speed $s_{t,v}$ (reward) of the minitaur. We consider continuous, discrete across-episode and discrete within-episode changes for the dynamics, and across-episode changes for the reward. We describe the settings in Appendix C.3.

**Baselines.** We compare our approach with a meta-RL approach for stationary RL, **VariBAD** [26], a meta-RL approach for non-stationary RL, **TRIO** [5], as well as with two representative task embedding approaches, **LILAC** [7] and **ZeUS** [8]. The details on the meta-learning setups are given in Appendix D.4. We also compare with stationary RL method, **SAC** [19], which will be our lower-bound, and compare with an **oracle** agent that has access to the full information of non-stationarity (e.g., the wind forces) and can use it to learn a policy, which will be our upper-bound. For all baselines, we use SAC for policy learning. We compare with the baselines on *average return* and *the compactness of the latent space* in varying degrees and functional forms of non-stationarity.

**Experimental results.** Fig. 2(a)-(c) shows the smoothed curves of average return across timesteps in a subsection of Half-Cheetah, Sawyer-Reaching, and Minitaur experiments. Smoothed curves of other experiments are given in Appendix Fig. A2 and A3. For continuous changes, we only compare with LILAC [7] since other approaches are not applicable. We smooth the learning curves by uniformly selecting a few data points for readability. Table 1 shows the results on Sawyer-Peg experiments using raw pixels as input, based on the reported results in [8]. The learning curve is given in Appendix Fig. A11. For a fair comparison, we indicate the best and final average return for each baseline in Fig. A11. Full results of all experiments are given in Appendix C.4, including significance tests for Wilcoxon signed-rank test at $\alpha = 0.05$ showing that FANS-RL is significantly better than baselines. We also visualize the learned graph in Sawyer in Appendix Sec C.7, showing that FN-VAE recovers reasonable causal relations in this domain, where the true graph is unknown.

**Ablation studies.** We conduct ablation studies on each component of FANS-RL and report some key results in Fig. 2(d). The ablation studies verify the effectiveness of all components, including binary masks/structure, smoothness loss, sparsity loss, reward prediction or state prediction. The results show that *the largest gain is provided by the factored representation*, validating our original hypothesis, followed by state prediction. As expected, reward prediction is also important when there is a nonstationary reward, while smoothness is important for within-episode changes. The disentangled design of CF inference networks is valuable when there are changes on both dynamics

and reward functions. Full results are in Appendix C.5, showing that learning which state components are affected by change, i.e. estimating the binary masks $C^{\theta^s \to s}$ and $C^{\theta^r \to r}$, provides the largest gains.

**Visualization on the learned $\theta$.** To verify that the learned $\theta$ can capture the true change factors, we compute the pairwise distance between learned $\theta^r$ at different time steps. We randomly sample 10 time steps from the Half-Cheetah experiment with across-episode changes on reward functions. Fig. 2(e) gives the pairwise distance of $\theta^r$ among 5 time steps (from episode 148, 279, 155, 159, 230) with different target speed values (0.04, 0.48, 0.89, 2.06, 2.85), respectively. We can find that there is a positive correlation between the distance of learned $\theta^r$ and values of change factors. Meanwhile, we also sample 5 time steps from episodes 31, 188, 234, 408 with target values around 1.67. Fig. 2(f) shows that the distance of $\theta^r$ among these 5 time steps is very small, indicating that the learned $\theta^r$ are almost the same for the similar values of change factors at different time steps. The visualization suggests that the learned $\theta$ can capture meaningful mappings from the time-varying factors.

**Varying latent dimensions, non-stationary levels and functional forms.** Fig. 2(g) shows the normalized averaged return versus the number of two latent features on Half-Cheetah. Our framework can learn a better policy with relatively smaller feature dimensions in the latent space than other approaches. As we show in Appendix C.4, Saywer and Minitaur have the similar trend, where we learn a better policy than the baselines with fewer latent features. We also vary the non-stationary levels in Half-Cheetah with discrete across-episode changes on both dynamics and rewards. A higher non-stationary degree indicates a faster change rate. Fig. 2(h) shows that FANS-RL achieves the highest return across all tested non-stationary degrees and that the gap increases with the non-stationarity. We also test FANS-RL together with all baselines on different non-stationary functions, including piecewise linear, damping-like and sinusoid waves. Fig. 2(i) displays the results, which indicate that FANS-RL can generally outperform the baselines on diverse non-stationary function forms. Detailed function equations and experimental setups can be referred to Appendix C.

**Multiple change factors** We consider different numbers and types of changes to verify the benefits from the factored structure in FANS-RL. We conduct experiments with 1) only change wind forces (1D); 2) change wind forces and gravity concurrently (2D); 3) change wind force and target speed (1D+1R); and 4) change wind force, gravity, and target speed together (2D+1R) in an across-episode way in Half-Cheetah. From Fig. 2(j), we find that, thanks to the factored representation, FANS-RL performs better in those more complicated scenarios with multiple numbers and types of changes.

## 6 Related Work

**Non-stationary and transfer RL.** Early works in non-stationary and transfer RL [27, 28] only detect changes that have already happened instead of anticipating them. If the evolution of non-stationary environments is a (Semi-)Markov chain, one can deal with non-stationarity with HM-MDPs [29] or HS3MDPs [30]. Several methods learn to anticipate changes in non-stationary deep RL. Chandak et al. [31] propose to maximize future rewards without explicitly modeling non-stationary environments. MBCD [32] uses change-point detection to decide if the agent should learn a novel policy or reuse previously trained policies. Al-Shedivat et al. [4] extend MAML [6] for the non-stationary setting, but do not explicitly model the temporal changes. TRIO [5] tracks the non-stationarity by inferring the evolution of latent parameters, which captures the temporal change factors during the meta-testing phase. ReBAL and GrBAL [33] meta-train the dynamic prior, which adapts to the local contexts efficiently. However, these methods have to meta-train the model on a set of non-stationary tasks, which may not be accessible in real-world applications. Another line of research directly learns the latent representation to capture the non-stationary components. In particular, LILAC [7] and ZeUS [8] leverage latent variable models to directly model the change factors in environments, and Guo et al. [34] estimate latent vectors that describe the non-stationary or variable part of the dynamics. Perez et al. [35, 36] estimate low-dimensional latent vectors that capture the changes in dynamics and rewards and leverage these inferred change factors to facilitate transfer RL across tasks. Similarly, François-Lavet et al. [37] use a compact state abstraction for recovering the sufficient low-dimensional representation of the environment, and only retrain or update some of its abstracted states for transferring to a novel but related task. Our approach fits in this line of work; however, as opposed to these methods, which model changes using a shared embedding space or a mixture model, and cannot distinguish which state components are affected, we use a factored representation.

**Factored MDPs.** Among works on factored representations in stationary settings, Hallak et al. [38] learn factored MDPs [11, 12, 13] to improve sample efficiency in model-based off-policy RL. Balaji

et al. [39] employ known factored MDP to improve both model-free and model-based RL algorithms. Working memory graphs [40] learn the factored observation space using Transformers. NeverNet [41] factorizes the state-action space through graph neural networks. Zholus et al. [42] factorize the visual states into actor-related, object of manipulation, and the latent influence factor between these two states to achieve sparse interaction in robotic control tasks. Differently from methods modeling the factored dynamics and rewards only, Zhou et al. [43] and Tang et al. [44] explore factored entities and actions, respectively. Zhou et al. [43] extend the factored MDP to the multi-entity environments, learning the compositional structure in tasks with multiple entities involved. Tang et al. [44] leverage the factored action space to improve the sample efficiency in healthcare applications. However, the factored structures in these two works are derived from inductive bias or domain experts.

**Factored MDPs, causality and multiple environments.** Several works leverage factored MDPs to improve the sample efficiency and generalization of RL. Most of these works focus on learning an invariant (e.g. causal) representation that fits all environments and do not support learning latent change factors. For example, AFaR [45] learns factored value functions via attention modules to improve sample efficiency and generalization across tasks. Mutti et al. [46] learn a causal structure that can generalize across a family of MDPs under different environments, assuming that there are no latent causal factors. Similarly, Wang et al. [47] propose to learn the factored and causal dynamics in model-based RL, in which the learned causal structure is assumed to generalize to unseen states. By deriving the state abstraction based on the causal graph, it can improve both the sample efficiency and generalizability of policy learning of MBRL. While most of the previous works focus on learning a causal structure that is time-invariant, Pitis et al. [48] learn a locally causal dynamics that can vary at each timestep and use it to generate counterfactual dynamics transitions in RL. While the previously described methods focus on a domain or time-invariant representations, in our work we also focus on modelling domain or time-specific factors in the form of latent change factors. A related work, AdaRL [14] learns the factored representation and the model change factors under heterogeneous domains with varying dynamics or reward functions. However, AdaRL is designed only for the domain adaptation setting and constant change factors without considering non-stationarity.

**Independent causal mechanisms.** Another related line of work is based on independent causal mechanisms [49, 50]. Recurrent independent mechanisms (RIMs) [51] learn the independent transition dynamics in RL with sparse communication among the latent states. Meta-RIMs [52] leverage meta-learning and soft attention to learn a set of RIMs with competition and communication. As opposed to these works, we do not assume that the mechanisms are independent and we learn the factored structure among all components in MDPs with a DBN.

## 7 Conclusions, Limitations and Future Work

We describe Factored Adaptation for Non-Stationary RL (FANS-RL), a framework that learns a factored representation for non-stationarity that can be combined with any RL algorithm. We formalize our problem as a Factored Non-stationary MDP (FN-MDP), augmenting a factored MDP with latent change factors evolving as a Markov process. FN-MDPs do not model a family of MDPs, but instead include the dynamics of change factors analogously to the dynamics of the states. This allows us to capture different non-stationarities, e.g., continuous and discrete changes, both within and across different episodes. To learn FN-MDPs we propose FN-VAEs, which we integrated in FANS-RL, an online model estimation and policy evaluation approach. We evaluate FANS-RL on benchmarks for continuous control and robotic manipulation, also with pixel inputs, and show it outperforms the state of the art on rewards and robustness to varying degrees of non-stationarity. Learning the graph in model estimation is computationally expensive, which limits the scalability of our approach. In future work, we plan to meta-learn the graphs among different tasks to improve the scalability of our approach and its applicability to complex RL problems, e.g., multi-agent RL.

**Acknowledgments**

FF would like to acknowledge the CityU High-Performance Computing (HPC) resources in Hong Kong SAR and LISA HPC from the SURF.nl. SM was supported by the MIT-IBM Watson AI Lab and the Air Force Office of Scientific Research under award number FA8655-22-1-7155. BH would like to acknowledge the support of Apple Scholarship. KZ was partially supported by the National Institutes of Health (NIH) under Contract R01HL159805, by the NSF-Convergence Accelerator Track-D award #2134901, by a grant from Apple Inc., and by a grant from KDDI Research Inc.

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
