# Appendix for Factored Adaptation for Non-Stationary Reinforcement Learning

## Contents

# A  Broader Impact

Our work is a first exploration in leveraging factored representations for non-stationary RL in order to improve adaptation. One of the limitations of our current approach, and similarly to a few other non-stationary RL approaches, is that we do not provide theoretical guarantees in terms of adapting to non-stationarity. This limits the applicability of these approaches in safety-critical applications, e.g. self-driving cars, or in adversarial environment. One of the future directions of this work would be to provide theoretical guarantees under reasonable assumptions, similarly to generalization bounds for factored representations for fast domain adaptation [14].

Based on the application, there might be different assumptions or inductive biases that might be considered reasonable. Since our work leverages insights from recent causality literature [9], we also inherit the same inductive bias in terms of assuming there is an underlying causal structure that is time-invariant throughout the non-stationarity. In our current method, this causal structure is estimated in the model estimation phase as one of the first steps of the algorithm. After this estimation, there can still be changes in the functional dependencies between the various components, but we assume there are no new edges/causal relations between components that were previously disconnected, or new forms of non-stationarity, in terms of connections between the change factor components and the state dimensions or reward. For example, if we consider Halfcheetah-v3 with a change in gravity and estimate a model that can handle this type of non-stationarity, our method will not be able to perform well under a new type of non-stationarity (e.g. change of wind forces) that was not observed during model estimation. An interesting extension of our work would be designing ways to efficiently detect changes in the causal structure and adapt the model.

As is the case with other works in causal discovery, we also make some standard assumptions to recover the causal graph from time series observational data. In particular, we assume that there are no other unobserved confounders, except for the change factors, and that there are no instantaneous causal effects between the state components, which is implied by our definition of an FN-MDP and its Dynamic Bayesian Network. In practical applications, this means that we are able to measure all the relevant causal variables and we are measuring them at a rate that is faster than their interaction. Additionally, in our identifiability proofs, we assume the causal Markov and faithfulness assumptions [53], which provide a correspondence between conditional independences and d-separations in the graph. The faithfulness assumption can be violated for example in case of deterministic relations, thus requiring a careful modelling of the system. In general, if these assumptions are violated, the causal structure we learn might be incorrect, and therefore the factored representation might not be beneficial to adapt to non-stationarity. A future direction would be to relax some of the current assumptions to provide a more realistic and flexible framework for factored non-stationary RL.

# B  Proofs and Causality Background

## B.1  Preliminaries

### B.1.1  Dynamic Bayesian networks

Dynamic Bayesian networks (DBNs) [54] are the extensions of Bayesian networks (BN), which model the time-dependent relationship between nodes (See an example in Fig. A1(b)). The unfolded DBNs can be represented as BNs. The variables in DBNs are in discrete time slices and dependent on variables from the same and previous time slices. Hence, the DBNs can model the stationary process repeated over the discrete time slices.

### B.1.2  Markov and faithfulness assumptions

Given a directed acyclic graph $G = (\mathbf{V}, \mathbf{E})$, where $\mathbf{V}$ is the set of nodes and $\mathbf{E}$ is the set of directed edges, we can define a graphical criterion that expresses a set of conditions on the paths.

**Definition 2** (d-separation [55]). *A path $p$ is said to be blocked by a set of nodes $\mathbf{Z} \subseteq \mathbf{V}$ if and only if (1) $p$ contains a chain $i \rightarrow m \rightarrow j$ or a fork $i \leftarrow m \rightarrow j$ such that the middle node $m$ is in $Z$, or (2) $p$ contains a collider $i \rightarrow m \leftarrow j$ such that the middle node $m$ is not in $\mathbf{Z}$ and such that no descendant of $m$ is in $\mathbf{Z}$. Let $X$, $Y$, and $Z$ be disjunct sets of nodes. $\mathbf{Z}$ is said to d-separate $X$ from $Y$ (denoted as $X \perp_d Y | \mathbf{Z}$) if and only if $\mathbf{Z}$ blocks every path from a node in $X$ to a node in $Y$.*

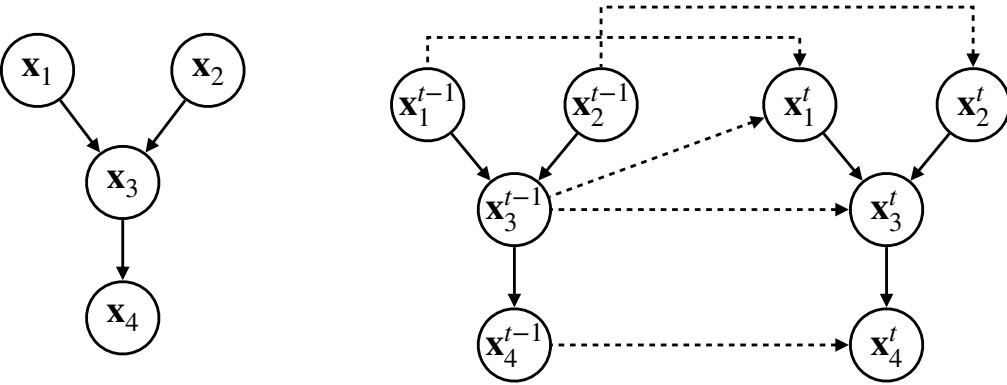

Figure A1: Examples on Bayesian and Dynamic Bayesian networks. The dashed edges indicate dependencies across time slices.

**Definition 3** (Global Markov Condition [53, 55])**.** *A distribution $P$ over $\mathbf{V}$ satisfies the global Markov condition on graph $G$ if for any partition $(\mathbf{X}, \mathbf{Z}, \mathbf{Y})$ such that $\mathbf{X}$ is d-separated from $\mathbf{Y}$ given $\mathbf{Z}$, i.e. $\mathbf{X} \perp_d \mathbf{Y}|\mathbf{Z}$ the distribution factorizes as:*

$$P(\mathbf{X}, \mathbf{Y}|\mathbf{Z}) = P(\mathbf{X}|\mathbf{Z})P(\mathbf{Y}|\mathbf{Z}).$$

*In other words, $\mathbf{X}$ is conditionally independent of $\mathbf{Y}$ given $\mathbf{Z}$, which we denote as $\mathbf{X} \perp\!\!\!\perp \mathbf{Y}|\mathbf{Z}$.*

**Definition 4** (Faithfulness Assumption [53, 55])**.** *There are no independencies between variables that are not entailed by the Markov Condition.*

If we assume both of these assumptions, then we can use d-separation as a criterion to read all of the conditional independences from a given DAG $G$. In particular, for any disjoint subset of nodes $\mathbf{X}, \mathbf{Y}, \mathbf{Z} \subseteq \mathbf{V}$: $\mathbf{X} \perp\!\!\!\perp \mathbf{Y}|\mathbf{Z} \iff \mathbf{X} \perp_d \mathbf{Y}|\mathbf{Z}$.

## B.2 AdaRL summary

Our work extends the factored representation for fast policy adaptation across domains introduced in AdaRL [14], which we summarize here. While Huang et al. [14] propose a general framework that can be applied to both MDPs and POMDPs, in this work we focus on MDPs, so we only present the simplified version of AdaRL for MDPs. The simplified AdaRL setting considers $n$ source domains and $n'$ target domains. The state at time $t$ is represented as $\mathbf{s}_t = (s_{1,t}, \cdots, s_{d,t})^\top \in \mathcal{S}^d$, while $\mathbf{a}_t \in \mathcal{A}^m$ is the executed action and $r_t \in \mathcal{R}$ is the reward signal. The generative process of the environment in the $k$-th domain with $k = 1, ..., n + n'$ can be described in terms of the transition function for each dimension $i = 1, ..., d$ of $\mathbf{s}_t$ as:

$$s_{i,t} = f_i(\boldsymbol{c}_i^{\boldsymbol{s} \to \boldsymbol{s}} \odot \mathbf{s}_{t-1}, \boldsymbol{c}_i^{\boldsymbol{a} \to \boldsymbol{s}} \odot \mathbf{a}_{t-1}, \boldsymbol{c}_i^{\boldsymbol{\theta}_k \to \boldsymbol{s}} \odot \boldsymbol{\theta}_k^{\boldsymbol{s}}, \epsilon_{i,t}^s) \tag{A1}$$

where $\odot$ denotes the element-wise product. The binary mask $\boldsymbol{c}_i^{\boldsymbol{s} \to \boldsymbol{s}} \in \{0,1\}^d$ represents which of the state components $s_{j,t-1}$ are used in the transition function of $s_{i,t}$. Similarly, $\boldsymbol{c}_i^{\boldsymbol{a} \to \boldsymbol{s}} \in \{0,1\}^m$ is a mask that indicates whether the action directly affects $s_{i,t}$. The change factor $\boldsymbol{\theta}_k^{\boldsymbol{s}} \in \mathbb{R}^p$ is the only parameter that depends on the domain $k$ in Eq. 1 and it encodes any change across domains in the dynamics. The binary mask $\boldsymbol{c}_i^{\boldsymbol{\theta}_k \to \boldsymbol{s}} \in \{0,1\}^p$ represents which of the $\boldsymbol{\theta}_k^{\boldsymbol{s}}$ components influence the $s_{i,t}$. Finally, $\epsilon_{i,t}^s$ is an i.i.d. random noise. Similarly the reward function is modeled as:

$$r_t = h(\boldsymbol{c}^{\boldsymbol{s} \to \boldsymbol{r}} \odot \mathbf{s}_{t-1}, \boldsymbol{c}^{\boldsymbol{a} \to \boldsymbol{r}} \odot \mathbf{a}_{t-1}, \boldsymbol{\theta}_k^r, \epsilon_t^r) \tag{A2}$$

where $\boldsymbol{c}_i^{\boldsymbol{s} \to \boldsymbol{r}} \in \{0,1\}^d$, $\boldsymbol{c}_i^{\boldsymbol{a} \to \boldsymbol{s}} \in \{0,1\}^m$, and $\epsilon_t^r$ is an i.i.d. random noise. The change factor $\boldsymbol{\theta}_k^r \in \mathbb{R}^q$ is the only parameter that depends on the domain $k$ in Eq. A2 and it encodes any change in the reward function. In this simplified setting, the binary masks $\boldsymbol{c}^{\to \cdot}$ can be seen as indicators of edges in a

Dynamic Bayesian Network (DBN). Under Markov and faithfulness assumptions, i.e., assuming the conditional independences in the data and d-separations in the true underlying graph coincide, the edges in the graph can be uniquely identified. This means one can learn the true causal graph representing jointly all of the environments, even if the change parameters are latent.

In the general AdaRL framework, the representation is learned via a combination of a state prediction network (to estimate the various $f_i$) and a reward prediction network (to estimate $h$). All binary masks $c^{\rightarrow \cdot}$ and change factors $\theta_k$ are trainable parameters. All change factors $\theta_k$ are assumed to be constant in each domain $k$. If the inputs are pixels, another encoder is added to infer the symbolic states, forming a *Multi-model Structured Sequential Variational Auto-Encoder (MiSS-VAE)*. MiSS-VAE leverages the generative modeling to learn the data generation process in RL system with multiple domains. All change factors in MiSS-VAE are modeled by constants in each domain. This setting is not suitable for non-stationary RL where the change factors evolve over time.

In general, not all of the dimensions of the learned state and change factor vectors are useful in policy learning. Huang et al. [14] select a subsection of dimensions which are essential for policy optimization. Leveraging the learned representation as a DBN, we can select compact states and CFs as having a directed path to a reward:

$$s_{i,t} \in \boldsymbol{s}^{min} \iff s_{i,t} \to \dots \to r_{t+\tau} \text{ for } \tau \geq 1$$
$$\theta_i \in \boldsymbol{\theta}^{min} \iff \theta_i \to \dots \to r_{t+\tau} \text{ for } \tau \geq 1$$

## B.3   Proofs

**Proposition 3** (Identifiability with observed change factors)**.** *Suppose all the change factors $\theta_t^s$ and $\theta_t^r$ are observed, i.e., Eq. 1-3) is an MDP. Under the Markov and faithfulness assumptions, all the binary masks $\boldsymbol{C}^{\cdot \to \cdot}$ are identifiable.*

*Proof.* We construct the graph in the Factored Non-stationary MDP as a dynamic Bayesian network (DBN) $G$ over the variables $\mathbf{V_{MDP}} = \{s_{1,t-1}, \dots, s_{d,t-1}, s_{1,t}, \dots, s_{d,t}, a_{1,t-1}, \dots, a_{m,t-1}, r_{t-1}\}$, and the change factors $\mathbf{V}_\theta = \{\theta_{1,t-1}^s, \dots, \theta_{p,t-1}^s, \theta_{1,t-1}^r, \dots, \theta_{q,t-1}^r, \theta_{1,t}^s, \dots, \theta_{p,t}^s, \theta_{1,t}^r, \dots, \theta_{q,t}^r\}$. In this setting, we can always the correct causal graph under the Markov and faithfulness assumptions. We rewrite the time index of $r_{t-1}$ as $r_t$.

We could rewrite also the change factors time indices by shifting them back in time by one step, i.e. $t \to t-1$, and this would allow us to model the whole setup as a Markov DBN without any instantaneous effect. In this setup, we can leverage existing results to show that the true causal graph is asymptotically identifiable from conditional independences in a time-series without any unobserved confounders or instantaneous effects [56]. In order to make our proof clearer, we instead show step by step how we can recover the parts of the graph related to $\mathbf{V_{MDP}}$, to $\mathbf{V}_\theta$ and finally the connections between them.

In this setting, there are no instantaneous effects except for the change factors, and the only causal parents for a variable that is not a change factor at time $t$ can be in the previous time-step $t-1$. In particular, as in usual MDPs, these are the only allowed edges:

1. state dimension $s_{i,t-1}$ at time $t-1$ to state dimension $s_{j,t}$ at time $t$, for $i,j \in \{1, \dots, d\}$ (this includes the case in which $i = j$);
2. action dimension $a_{k,t-1}$ at time $t-1$ to state dimension $a_{j,t}$ at time $t$, for $j \in \{1, \dots, d\}, k \in \{1, \dots, m\}$;
3. state dimension $s_{i,t-1}$ at time $t-1$ to reward $r_t$ at time $t$, for $i \in \{1, \dots, d\}, k \in \{1, \dots, m\}$;
4. action dimension $a_{k,t-1}$ at time $t-1$ to reward $r_t$ at time $t$, for $k \in \{1, \dots, m\}$;

and in addition we have some extra knowledge about the allowed edges to and from change factors, as expressed in our generative model in Equations (1-3):

1. transition change factor dimension $\theta_{i,t}^s$ at time $t$ to state dimension $s_{j,t}$ at time $t$, for $i \in \{1, \dots, p\}, j \in \{1, \dots, d\}$;
2. transition change factor dimension $\theta_{i,t}^s$ at time $t$ to transition change factor dimension $\theta_{j,t+1}^s$ at time $t+1$, for $i,j \in \{1, \dots, p\}$;

3. reward change factor dimension $\theta^r_{i,t}$ at time $t$ to reward change factor dimension $\theta^r_{j,t+1}$ at time $t+1$, for $i, j \in \{1, \ldots, q\}$;

For the reward change factors, we assume they are fully connected to the reward, as shown in Eq. (2). Using this background knowledge, we can learn the edges from any other variable $V_{i,t-1} \to V_{j,t}$, for $V_{i,t-1}, V_{j,t} \in \mathbf{V_{MDP}}$, just by checking if $V_{i,t-1} \not\!\perp\!\!\!\perp V_{i,t}|\theta^r_t, \theta^s_t, \mathbf{s}_{t-2}, \mathbf{a}_{t-2}$. This dependence implies that the variables are d-connected, under the Markov and faithfulness assumptions. Except for a direct edge, there is no other possible path through the graph unrolled in time, since we have blocked all influence of time-step $t-2$ and earlier, and the paths through future time-steps contain colliders. So this means that $V_{i,t-1}$ and $V_{j,t}$ are adjacent, and in particular the edge follows the arrow of time, from $t-1$ to $t$. In our setting, we do not model the actions at time $t$, since we assume they are not caused by any other variable. We also never need to condition on the reward to check if two variables are adjacent, since it's always on a collider path. This means we are able to learn the following binary masks:

1. state dimensions to state dimensions $\boldsymbol{C^{s \to s}}$;
2. action dimensions to state dimensions $\boldsymbol{C^{a \to s}}$;
3. state dimensions to reward $\boldsymbol{c^{s \to r}}$
4. action dimensions to reward $\boldsymbol{c^{a \to r}}$;

To learn the edges between any change factor component to another change factor component, i.e., $V_{i,t-1} \to V_{j,t}$ for $V_{i,t-1}, V_{j,t} \in \mathbf{V}_\theta$, we can just check if $V_{i,t-1} \not\!\perp\!\!\!\perp V_{i,t}|\theta^r_{t-2}, \theta^s_{t-2}$, since states, actions and rewards can never be parents of the change factors, so we do not need to condition on them to close any path through the earlier time-steps. We also assumed that change factor components follow a Markov process, so they do not have instantaneous effects towards each other. This means we are able to learn the following masks:

1. transition change factor dimensions to transition change factor dimensions $\boldsymbol{C^{\theta^s \to \theta^s}}$;
2. reward change factor dimensions to reward change factor dimensions $\boldsymbol{C^{\theta^r \to \theta^r}}$;

Finally to learn the edges between the change factors $V_{i,t} \in \mathbf{V}_\theta$ and the other variables $V_{j,t} \in \mathbf{V_{MDP}}$, we can just check if $V_{i,t} \not\!\perp\!\!\!\perp V_{j,t}|\mathbf{s}_{t-1}, \mathbf{a}_{t-1}, \theta^r_{t-1}, \theta^s_{t-1}$ and if this is true we can learn the edge $V_{i,t} \to V_{j,t}$. This means we are able to learn the mask:

1. transition change factor dimensions to state dimensions $\boldsymbol{C^{\theta^s \to s}}$;

All of these results together show that if we have the Markov and faithfulness assumption, i.e. the conditional independence tests return the true d-separations in the graph, and we observe the change factors, we are able to completely identify the graph of the FN-MDP represented by the binary masks. $\qquad \square$

**Proposition 4** (Partial Identifiability with latent change factors). *Suppose the generative process follows Eq. 1-3 and the change factors $\boldsymbol{\theta^s_t}$ and $\boldsymbol{\theta^r_t}$ are unobserved. Under the Markov and faithfulness assumptions, the binary masks $\boldsymbol{C^{s \to s}}, \boldsymbol{C^{a \to s}}, \boldsymbol{c^{s \to r}}$ and $\boldsymbol{c^{a \to r}}$. Moreover, we can identify which state dimensions are affected by $\boldsymbol{\theta^s_t}$ and if the reward function changes.*

*Proof.* In this case, the MDP is non-stationary, since we cannot observe the latent change factors. We assume that we can represent the latent change factors as a smooth function of the observed time index $t$. This assumption is called *pseudo-causal sufficiency* in previous work [9]. We can then use the time index $t$ as a surrogate variable to characterize the unobserved change factors, since at each time-step their value will be fixed.

We again consider a DBN $\mathcal{G}_{MDP \cup \{t\}}$ over the variables $\mathbf{V_{MDP}} = \{s_{1,t-1}, \ldots, s_{d,t-1}, s_{1,t}, \ldots, s_{d,t}, a_{1,t-1}, \ldots, a_{m,t-1}, r_{t-1}\}$ and the time index $t$. We rewrite the time index of $r_{t-1}$ as $r_t$. Note that we do not represent the change factors in this DBN, but we can capture their effect through $t$ since they are assumed to be deterministic smooth functions of the time index.

We can then reuse the results by Huang et al. [9] (Theorem 1) in which under the pseudo-causal sufficiency (Assumption 1 in that paper) and the Markov and faithfulness assumption (Assumption 2), one can asymptotically identify the true causal skeleton (i.e. the adjacencies) in the graph $\mathcal{G}_{MDP \cup \{t\}}$ through conditional independence tests. In particular for any $V_i, V_j \in \mathbf{V}_{MDP} \cup \{t\}$, $V_i$ and $V_j$ are not adjacent if there exists a subset of the unrolled graph $\mathbf{V}_k$ of $\mathbf{V}_{MDP \cup \{t\}} \setminus \{V_i, V_j\}$ in $\mathcal{G}_{MDP \cup \{t\}}$

such that $V_i \perp\!\!\!\perp V_j | \mathbf{V}_k$. In particular, we can also focus on the tests that were used for the proof of the previous Proposition, by just substituting the change factors with the time index $t$. The skeleton is the undirected version of the causal graph, so this result only tells us that can asymptotically get the correct undirected edges, but not their orientations.

Fortunately, in our setting we have some additional background knowledge that allows us to orient all the existing edges. In particular, as in usual MDPs, these are the only allowed edges:

1. state dimension $s_{i,t-1}$ at time $t-1$ to state dimension $s_{j,t}$ at time $t$, for $i, j \in \{1, \dots, d\}$ (this includes the case in which $i = j$);
2. action dimension $a_{k,t-1}$ at time $t-1$ to state dimension $a_{j,t}$ at time $t$, for $j \in \{1, \dots, d\}, k \in \{1, \dots, m\}$;
3. state dimension $s_{i,t-1}$ at time $t-1$ to reward $r_t$ at time $t$, for $i \in \{1, \dots, d\}, k \in \{1, \dots, m\}$;
4. action dimension $a_{k,t-1}$ at time $t-1$ to reward $r_t$ at time $t$, for $k \in \{1, \dots, m\}$;

This means that, for example, we cannot have a variable at time $t$ causing a variable at time $t-1$. Therefore if two variables $V_{i,t-1}$ and $V_{j,t}$ are adjacent, we already know that the direction of that edge will be $V_{i,t-1} \to V_{j,t}$. This also implies that the following binary masks are identifiable (i.e. no edge remains unoriented):

1. state dimensions to state dimensions $\boldsymbol{C^{s \to s}}$;
2. action dimensions to state dimensions $\boldsymbol{C^{a \to s}}$;
3. state dimensions to reward $\boldsymbol{c^{s \to r}}$
4. action dimensions to reward $\boldsymbol{c^{a \to r}}$;

These represent all of the edges in $\mathcal{G}_{MDP}$. We can also learn the edges from $t$ to $V_{MDP}$ (by construction we assume the opposite direction is not possible), which will represent the effect of the change factors, as we show in the following.

Since $t$ inherits all of the children of the latent change factors in $G$, we can further show that if $s_{i,t} \perp\!\!\!\perp t | \boldsymbol{s}_{t-1}, \boldsymbol{a}_{t-1}$ in $G$, then none of the latent change factor dimensions $\theta_{j,t}^s$ affect $s_{i,t}$, i.e., $s_{i,t} \perp\!\!\!\perp t | \boldsymbol{s}_{t-1}, \boldsymbol{a}_{t-1} \iff c_{i,j}^{\theta^s \to s} = 0$. Intuitively, this means that the distribution of $s_{i,t}$ only depends on $\mathbf{s}_{t-1}$ and $\mathbf{a}_{t-1}$, and not on the timestep $t$, or in other words, this distribution is stationary. Under the same principle, if $r_t \perp\!\!\!\perp t | \boldsymbol{s}_t, \boldsymbol{a}_t$, then the reward is stationary.

$\square$

## C  Details on Experimental Designs and Results

### C.1  MuJoCo

We modify the Half-Cheetah environment into a variety of non-stationary settings. Details on the change factors are given as below.

**Changes on dynamics.**  We change the wind forces $f^w$ in the environment. We consider the changing functions can be both continuous and discrete.

• Continuous changes: $f_t^w = 10 + 10\sin(0.005 \cdot t)$, where $t$ is the timestep index;
• Discrete changes: (1) Across-episode:
a. Sine function: $f_w = 10 + 10\sin(0.5 \cdot i)$
b. Damping-like function: $f_w = 10 + 3 \cdot (1.01)^{-\lceil i/10 \rceil} \sin(0.5 \cdot i)$
c. Piecewise linear function $f_w = 5 + 0.02 \cdot \|i - 1500\|$; , where $i$ is the episode index.
(2) Within-episode: $f_w = 10 + 10\sin(0.4 \cdot \lfloor t/10 \rfloor)$, where $t$ is the timestep index.

We also consider a special case where the agent's mechanism is changing over time. Specifically, the one random joint is disabled at the beginning of each episode.

**Changes on reward functions.**  To introduce non-stationarity in the rewards, we change the target speed $v_g$ in each episode. To make the learning process stable, we only consider the discrete changes and the change points are located at the beginning of each episode. The changing function is $v_g = 1.5 + 1.5\sin(0.2 \cdot i)$, where $i$ denotes the episode index.

**Changes on both dynamics and rewards.** We consider a more general but challenging scenario, where the changes on dynamics and rewards can happen concurrently during the lifetime of the agents. We change the wind forces and target speed at the beginning of each episode. At episode $i$, the dynamics and reward functions are:

$$\begin{cases} f_w = 10 + 10\sin(w \cdot i) \\ v_g = 1.5 + 1.5\sin(w \cdot i) \end{cases}$$

Here, $w$ is the non-stationary degree. We consider multiple values of $w$ in our experiments. In Fig. 2(d), $w = 0.5$. In Fig. 2(h), $w$ is the value of non-stationary degree.

## C.2 Sawyer benchmarks

In Sawyer-Reaching, the sawyer arm is trained to reach a target position $s_t^g$. The reward $r_t$ is the difference between the current position $s_t$ and the target position $r_t = -\|s_t - s^g\|_2$. In this task, we cannot directly modify the dynamics in the simulator, so consider a reward-varying scenario, where the target location changes across each episode following a periodic function. In Sawyer-Peg, the robot arm is trained to insert a peg into a designed target location $s^g$. The reward function is $r_t = \mathbb{I}(\|s_t - s^g\|_2 \le 0.05)$.

We change the target location in Sawyer reaching task. The target location $s_t^g$ is given as below:

$$\mathbf{s}_t^g = \begin{bmatrix} 0.1 \cdot \|\cos(0.2 \cdot i)\| \\ 0.1 \cdot \sin(0.5 \cdot i) \\ 0.2 \end{bmatrix}$$

where $i$ is the episode index. For Sawyer-Peg task, the target location $s_g$ changes at each episode. The parameters in each dimension of $s_g$ is randomly sampled at episode $i$ as below:

- x_range_1: $(0.44, 0.45)$;
- x_range_2: $(0.6, 0.61)$;
- y_range_1: $(-0.08, -0.07)$;
- y_range_2: $(0.07, 0.08)$;

## C.3 Minitaur benchmarks

We consider both the changes on dynamics and reward functions.

**Changes on dynamics.** We change the mass of taur $m$ in the environment. Specifically, we consider both the continuous and discrete changes.

- Continuous changes: $m_t = 1.0 + 0.75\sin(0.005 \cdot t)$;
- Discrete and within-episode changes: $m_t = 1.0 + 0.75\sin(0.3 \cdot \lfloor t/20 \rfloor)$

**Changes on both dynamics and reward functions.** We also consider a case where both the dynamics and reward functions change at the beginning of each episode. We change the target speed of minitaur to introduce the non-stationarity of reward functions. The change functions are given below:

$$\begin{cases} m_i = 1.0 + 0.5\sin(0.5 \cdot i) \\ s_v = 0.3 + 0.2\sin(0.5 \cdot i) \end{cases}$$

## C.4 Full results

Fig. A2 and A3 give the smoothed learning curves on average return over 10 runs versus timesteps in Half-Cheetah and Minitaur experiments. Table C.4 shows the average final return over 10 runs for all experiments. Fig. A4 demonstrates the return on Half-Cheetah with different non-stationary degrees on multi-factor changing scenario. Fig. A5 gives average return on different benchmarks with varying numbers of latent features with all evaluated approaches.

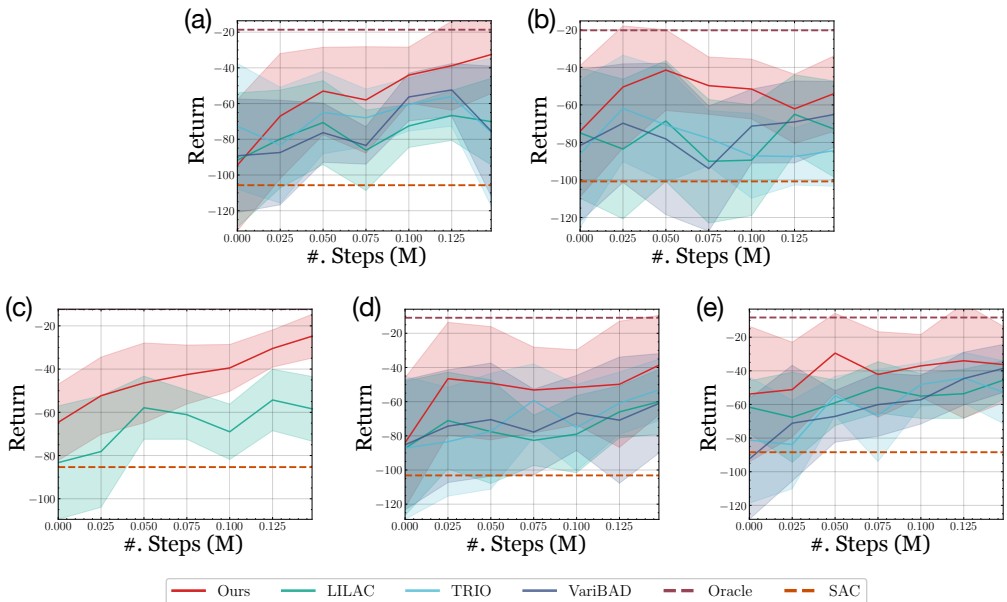

Figure A2: The average return (smoothed) across timesteps in Half-Cheetah experiments. (a) Discrete (across-episode) changes on wind forces; (b) Discrete (within-episode) changes on wind forces; (c) Continuous changes on wind forces; (d) Discrete (across-episode) changes on target speed; (e) Discrete (across-episode) changes on wind forces and target speed concurrently.

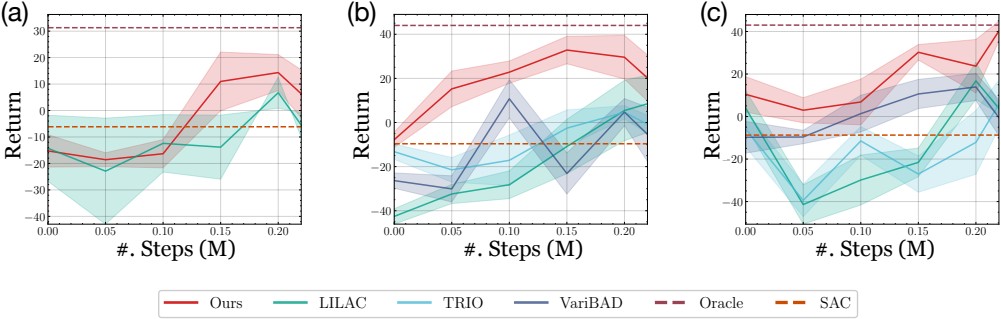

Figure A3: The average return (smoothed) across timesteps in Minitaur experiments. (a) Continuous changes on the mass; (b) Discrete (across-episode) changes on the target speed; (c) Discrete (across-episode) changes on mass and target speed concurrently.

|  | Oracle | SAC | LILAC | TRIO | VariBAD | Ours |
|---|---|---|---|---|---|---|
| Half-Cheetah: A-EP (D) | −24.4 (±16.2) | −113.4 • (±28.5) | −70.1 • (±27.7) | −76.0 • (±47.3) | −75.5 • (±41.6) | **-32.6** (±25.0) |
| Half-Cheetah: A-EP (A) | −9.6 (±5.7) | −30.5 • (±12.1) | −19.4 • (±11.4) | −21.9 • (±13.0) | −17.3 • (±10.2) | **-15.1** (±9.8) |
| Half-Cheetah: W-EP (D) | −48.2 (±41.6) | −107.5 • (±20.6) | −72.9 • (±29.3) | −84.4 • (±21.7) | −65.1 • (±20.1) | **-54.0** (±23.0) |
| Half-Cheetah: CONT (D) | −12.3 (±27.7) | −112.0 • (±16.9) | −58.4 • (±22.3) | - | - | **-24.8** (±21.1) |
| Half-Cheetah: A-EP (R) | −10.9 (±20.1) | −131.5 • (±16.9) | −60.1 • (±21.7) | −53.1 • (±20.6) | −61.0 • (±33.3) | **-38.7** (±33.3) |
| Half-Cheetah: A-EP (R+D) | −15.2 (±38.1) | −105.3 • (±38.1) | −45.6 (±13.1) | −52.6 • (±21.4) | −38.6 (±16.3) | **-36.2** (±26.0) |
| Sawyer-Reaching: A-EP (R) | 6.4 (±3.9) | −52.5 • (±9.1) | −34.0 • (±8.2) | −28.1 • (±2.9) | −31.3 • (±4.3) | **-9.7** (±2.5) |
| Minitaur: CONT (D) | 31.3 (±4.2) | −6.1 • (±3.9) | −5.5 • (±11.7) | - | - | **6.3** (±10.4) |
| Minitaur: W-EP (D) | 44.9 (±5.8) | −9.6 • (±5.5) | 8.5 • (±14.9) | −0.8 • (±4.7) | 5.4 • (±14.1) | **20.2** (±11.9) |
| Minitaur: A-EP (R+D) | 43.0 (±4.7) | −8.7 • (±5.4) | 3.8 • (±3.0) | 5.8 • (±12.9) | 21.5 • (±9.7) | **40.2** (±5.3) |

Table A1: Average final return of different methods on Half-Cheetah, Sawyer-Reaching, and minitaur benchmarks with a variety of non-stationary settings. The best non-oracle results w.r.t. the mean are marked in **bold**. "•" indicates the baseline for which the improvements of our approach are statistically significant (via Wilcoxon signed-rank test at $5\%$ significance level). D, R, and A denote changes on dynamics, reward and agent's mechanism respectively. A-EP, W-EP, and CONT denote across-episode, within-episode and continuous changes, respectively.

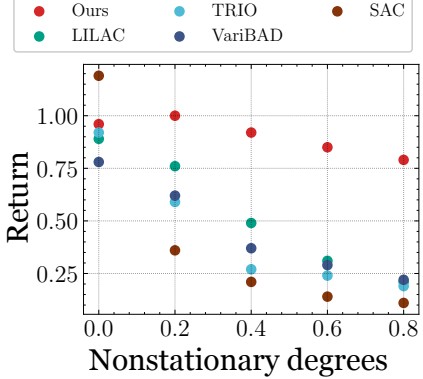

Figure A4: Average final return on 10 runs on Half-Cheetah with different non-stationary degrees on across-episode and multi-factor changes.

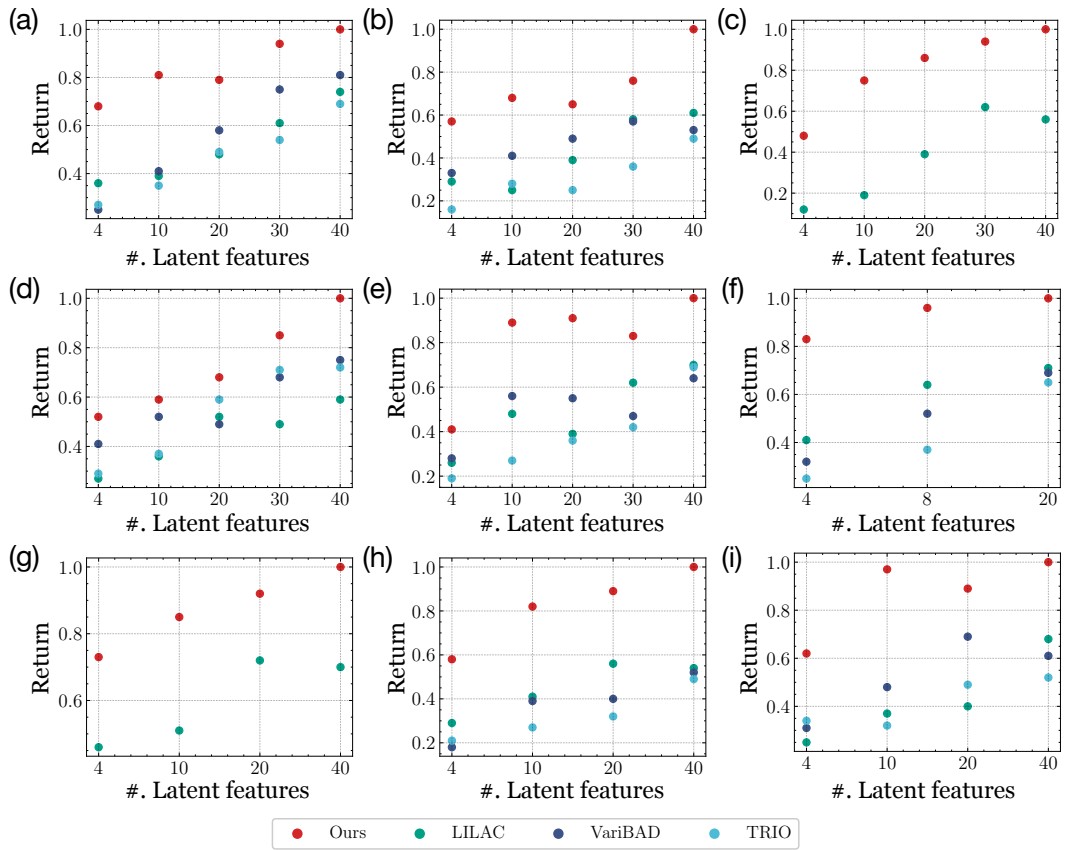

Figure A5: Average return on different benchmarks with different number of latent features. (a) Half-Cheetah experiments with discrete (across-episode) changes on wind forces; (b) Half-Cheetah experiments with discrete (within-episode) changes on wind forces; (c) Half-Cheetah experiments with continuous changes on wind forces; (d) Half-Cheetah experiments with discrete (across-episode) changes on target speed; (e) Half-Cheetah experiments with discrete (across-episode) changes on wind forces and target speed concurrently; (f) Sawyer-Reaching experiment with discrete (across-episode) changes on target locations; (g) Minitaur experiments with continuous changes on the mass; (h) Minitaur experiments with discrete (across-episode) changes on the target speed; (i) Minitaur experiments with discrete (across-episode) changes on mass and target speed concurrently.

## C.5 Ablation studies on FANS-RL

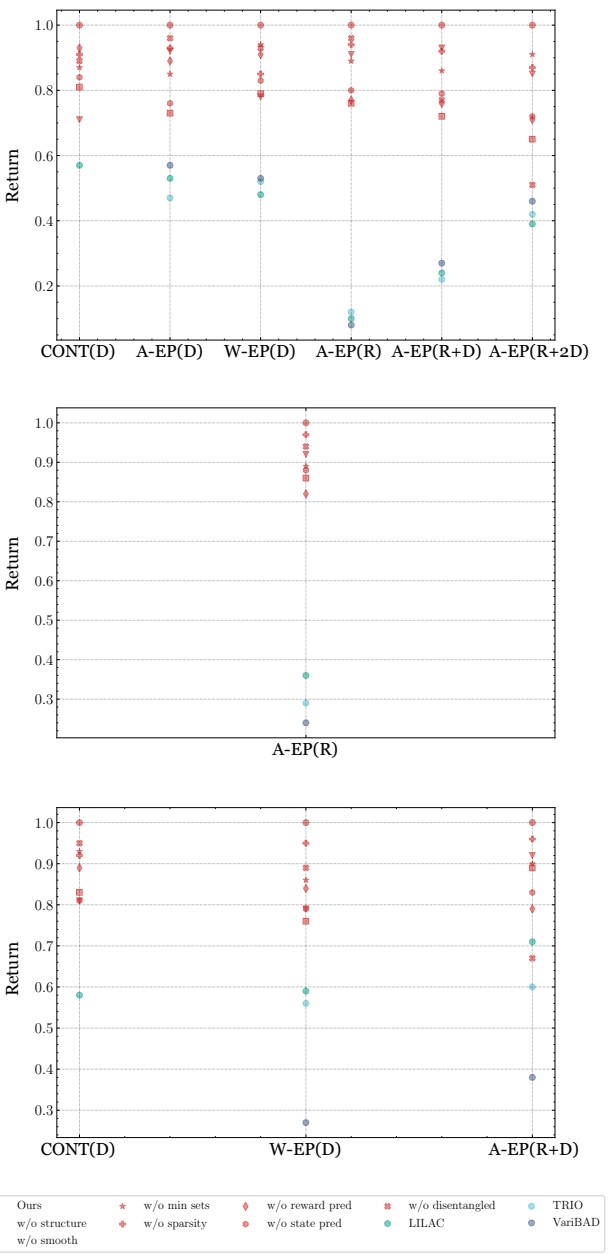

Figure A6: Ablation studies on different components in FANS-RL on (a) Half-Cheetah experiment; (b) Sawyer experiment; and (c) Minitaur experiments. CONT, A-, W-EP indicate continuous, across-episode, and within-episode changes, respectively. (D) and (R) represent changes on dynamics and reward functions, respectively. Best viewed in color.

To verify the effectiveness of each component in our proposed framework, we consider the following ablation studies:

- Without smoothness loss ($\mathcal{L}_{\text{smooth}}$);
- Without structural relationships ($C^{\cdot \rightarrow \cdot}$);
- Without compact representations ($s^{min}, \theta^{min}$);

- Without sparsity losses ($\mathcal{L}_{\text{sparse}}$);

- Without reward or state prediction losses ($\mathcal{L}_{\text{pred-rw}}$, $\mathcal{L}_{\text{pred-dyn}}$);

- Without the disentangled design of CF inference networks for dynamics ($q_\phi^s$) and rewards ($q_\phi^r$). Specifically, we use one CF inference encoder and the mixed latent space of $\boldsymbol{\theta}^s$ and $\boldsymbol{\theta}^r$ in this setting.

As shown in Fig. A6, all the studied components benefit the performance. Furthermore, FANS-RL can still outperform the strong baselines even without some of the components. Additionally, we conduct the ablation studies where $C^{s\to s}$, $C^{\theta^s\to s}$ (or $C^{\theta^r\to r}$), and $C^{\theta^s\to\theta^s}$ (or $C^{\theta^r\to\theta^r}$) have been switched off. The results in Table C.5 indicate that 1) all these binary masks could lead to better policy learning performances; and 2) $C^{\theta^s\to s}$ and $C^{\theta^r\to r}$ are the most crucial elements among them. This implies that the changes happen in a sparse manner and our model can capture the sparse changes in dynamics or reward functions.

| | Ours | w/o $C^{s\to s}$ | w/o $C^{\theta^s\to s}$ or $C^{\theta^r\to r}$ | w/o $C^{\theta^s\to\theta^s}$ or $C^{\theta^r\to\theta^r}$ |
|---|---|---|---|---|
| Half-Cheetah: A-EP (D) | 1.00 | 0.93 | 0.84 | 0.95 |
| Half-Cheetah: W-EP (D) | 1.00 | 0.89 | 0.80 | 0.86 |
| Half-Cheetah: CONT (D) | 1.00 | 0.92 | 0.87 | 0.94 |
| Half-Cheetah: A-EP (R) | 1.00 | 0.85 | 0.79 | 0.83 |
| Half-Cheetah: A-EP (R+D) | 1.00 | 0.83 | 0.75 | 0.89 |
| Sawyer-Reaching: A-EP (R) | 1.00 | 0.95 | 0.90 | 0.92 |
| Minitaur: CONT (D) | 1.00 | 0.96 | 0.85 | 0.98 |
| Minitaur: W-EP (D) | 1.00 | 0.86 | 0.78 | 0.92 |
| Minitaur: A-EP (R+D) | 1.00 | 0.94 | 0.89 | 0.88 |

Table A2: Average final return of our framework for the ablations in which we switch off the estimation of some binary masks.

We also test different smoothness losses, including the moving average (MA) $\mathcal{L}_{\text{smooth}} = \sum_{t=2}^{T}\left(\|\theta_t - (\theta_{t-1} + \theta_{t-2} + \ldots + \theta_{t-T})/T\|_1\right)$ and exponential moving average (EMA) $\mathcal{L}_{\text{smooth}} = \sum_{t=2}^{T}\left(\|\theta_t - (\beta\theta_{t-1} + (1-\beta)\mathbf{v}_{t-2})\|_1\right)$, where $\mathbf{v}_t = \beta\theta_t + (1-\beta)\mathbf{v}_{t-1}$ and $\mathbf{v}_0$ is a zero vector. Table C.5 shows the normalized final results of using different smoothness losses. We can find that different smoothness losses have comparable performances.

| | Ours | MA ($T=2$) | EMA ($\beta=0.98$) |
|---|---|---|---|
| Half-Cheetah: A-EP (D_1) | 1.00 | 1.02 | 0.89 |
| Half-Cheetah: A-EP (D_2) | 1.00 | 0.96 | 0.90 |
| Half-Cheetah: W-EP (D) | 1.00 | 0.88 | 1.05 |
| Half-Cheetah: CONT (D) | 1.00 | 1.04 | 0.95 |
| Half-Cheetah: A-EP (R) | 1.00 | 0.93 | 0.82 |
| Half-Cheetah: A-EP (R+D) | 1.00 | 1.09 | 1.02 |
| Sawyer-Reaching: A-EP (R) | 1.00 | 0.97 | 0.91 |
| Minitaur: CONT (D) | 1.00 | 1.08 | 0.96 |
| Minitaur: W-EP (D) | 1.00 | 0.86 | 1.03 |
| Minitaur: A-EP (R+D) | 1.00 | 0.97 | 0.94 |

Table A3: Average final return of using different smoothness losses.

## C.6 Visualization on the learned change factors

Fig. A7 gives the visualization on the learned $\theta$ in Half-Cheetah. Fig. A7(a-b) show the pairwise Euclidean distance between learned $\boldsymbol{\theta}^r$ and the axes denote the values of change factors on rewards. Similarly, Fig. A7(c-d) displays the Euclidean distance between learned $\boldsymbol{\theta}^s$ and the values of change factors on dynamics. The results suggest that there is a positive correlation between the distance of

learned $\boldsymbol{\theta}$ versus the true change factors. This can verify that $\boldsymbol{\theta}$ can capture the change factors in the system.

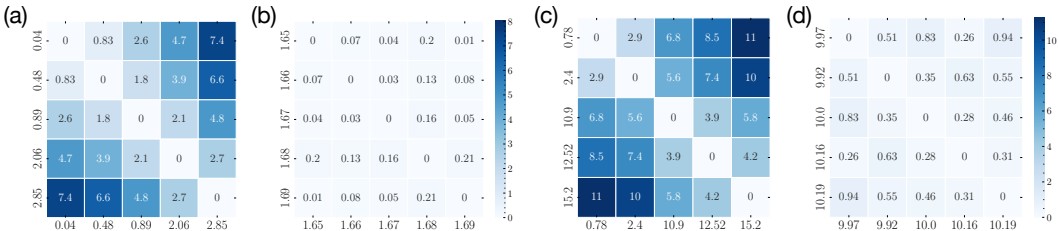

Figure A7: Visualization on the learned $\boldsymbol{\theta}^r$ and $\boldsymbol{\theta}^s$. Best viewed in color.

## C.7 Visualization on the learned graphs

In most of the enviroments we considered the underlying causal graph is unknown. Instead, we will focus on evaluating whether the learned causal graph encodes reasonable causal relations in the domain. Fig. A8 shows the learned graph in the Sawyer environment. $\mathbf{S}_1$, $\mathbf{S}_2$, and $\mathbf{S}_3$ are the state variables of the end-effector, gripper, and the manipulated object respectively. Here the change factor $\theta^r$ denotes the change in the reward function. These learned graphs represent a set of reasonable cause-effect relations in the physical system, in which 1) the gripper is affected by the end-effector; 2) the object can be affected by the gripper (e.g., pushing, picking, etc.); 3) the action manipulates the end-effector and the gripper; and 4) the reward is dependent on the object and the gripper.

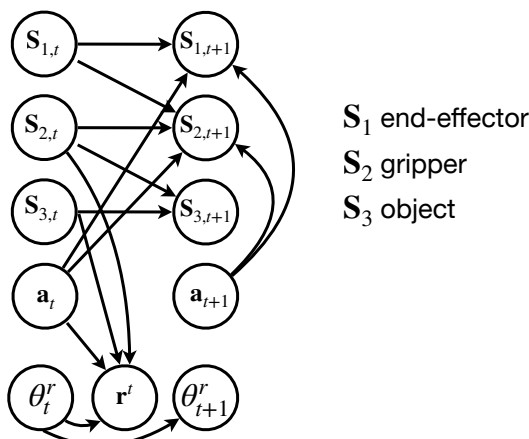

Figure A8: Visualization on the learned graph in Sawyer environment.

## D Details on the Factored Adaptation Framework

Fig. A9 gives the pipeline of the whole framework.

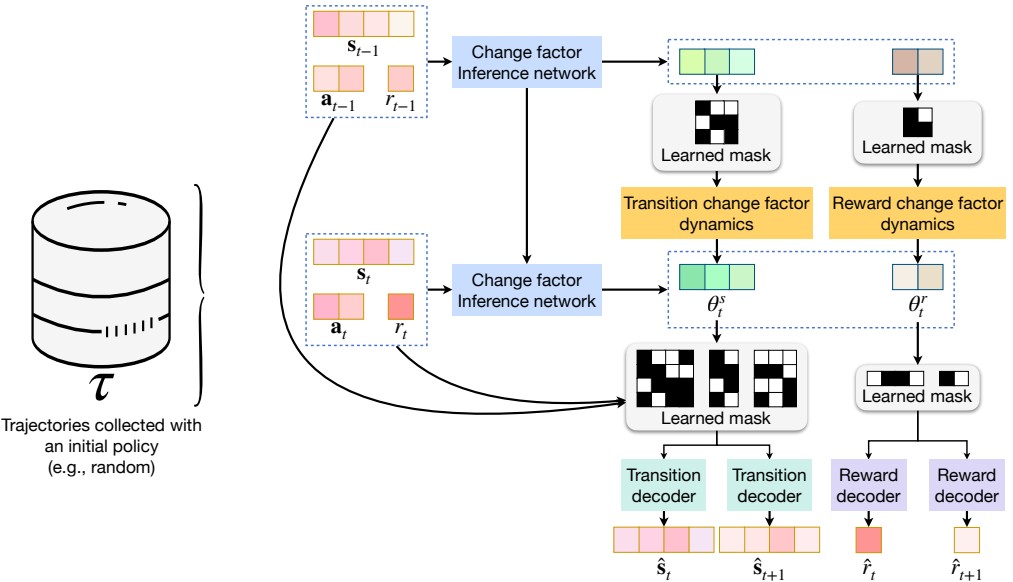

Figure A9: The whole pipeline of FANS-RL.

## D.1 Algorithm pipelines of FN-VAE

Alg. A1 gives the full pipeline of FN-VAE.

---

**Algorithm A1:** Learning FN-MDPs using FN-VAE.

---

**Input:** Trajectories $\tau$, FN-VAE parameters $\phi = (\phi^s, \phi^r)$, $\alpha = (\alpha_1, \alpha_2)$, $\beta = (\beta_1, \beta_2)$, $\gamma$; Mask matrices $G = (C^{\cdots})$, Boolean updateG, Learning rates $\lambda_\phi, \lambda_\gamma, \lambda_\alpha, \lambda_\beta, \lambda_G$, Length of collected rollouts $k$; Number of training epochs E.
**Output:** $\phi$, $\alpha_1$, $\alpha_2$, $\beta_1$, $\beta_2$, $\gamma$
**for** $i = 1, 2, \ldots,$ E **do**
    Randomly sample a batch of trajectories $\tau_{0:k}$ in $\tau$
    `# Infer the latent change factors`
    **for** j = s, r **do**
        Infer $\mu_{\phi^j}(\tau_{0:k})$ and $\sigma^2_{\phi^j}(\tau_{0:k})$ using $q_{\phi^j}$
        Infer $\mu_{\gamma^j}(\theta^j_{0:k})$ and $\sigma^2_{\gamma^j}(\theta^j_{0:k})$ using $p_{\gamma^j}$
        Sample $\theta^j_{0:k} \sim \mathcal{N}\left(\mu_{\phi^j}(\tau_{0:k}), \sigma^2_{\phi^j}(\tau_{0:k})\right)$
    **end for**
    Reconstruct and predict $\hat{s}_{0:k}, \hat{s}_{1:k}, \hat{r}_{0:k}, \hat{r}_{1:k}$ using $p_{\alpha_1}, p_{\alpha_2}, p_{\beta_1}$, and $p_{\beta_2}$
    `# Update the FN-VAE model`
    $\phi \leftarrow \phi - \lambda_\phi \nabla_\phi \mathcal{L}_{\text{VAE}}$
    $\gamma \leftarrow \gamma - \lambda_\gamma \nabla_\gamma (\mathcal{L}_{\text{KL}} + \mathcal{L}_{\text{smooth}})$
    $\alpha \leftarrow \alpha - \lambda_\alpha \nabla_\alpha (\mathcal{L}_{\text{rec-dyn}} + \mathcal{L}_{\text{pred-dyn}})$
    $\beta \leftarrow \beta - \lambda_\beta \nabla_\beta (\mathcal{L}_{\text{rec-rw}} + \mathcal{L}_{\text{pred-rw}})$
    **if** updateG **then**
        $G \leftarrow G - \lambda_G \nabla_G (\mathcal{L}_{\text{rec-dyn}} + \mathcal{L}_{\text{rec-rw}} + \mathcal{L}_{\text{KL}} + \mathcal{L}_{\text{sparse}})$
    **end if**
**end for**

---

## D.2 The framework dealing with discrete and across-episode changes

Alg. A2 gives the extended framework for handling both across- and within-episode changes in non-stationary RL, respectively. The major difference between Alg. A2 and Alg. 1 is that we

**Algorithm A2:** Factored Adaptation for non-stationary RL (discrete changes.)

---

1: **Init:** Env; VAE parameters: $\phi = (\phi^s, \phi^r)$, $\alpha = (\alpha_1, \alpha_2)$, $\beta = (\beta_1, \beta_2)$, $\gamma$; Mask matrices: $\boldsymbol{C}^{\cdot \rightarrow \cdot}$;
   Policy parameters: $\psi$; replay buffer: $\mathcal{D}$; Number of episodes: $N$; Episode horizon: $H$; Change
   index $\tilde{\boldsymbol{t}} = \{t_1, \ldots, t_M\}$; $m = 0$.
2: **Output:** $\phi, \alpha_1, \alpha_2, \beta_1, \beta_2, \gamma, \psi$
3: # Model initialization
4: Collect multiple trajectories $\boldsymbol{\tau} = \{\boldsymbol{\tau}_{0:k}^1, \boldsymbol{\tau}_{0:k}^2, \ldots\}$ with policy $\pi_\psi$ from Env;
5: Learn an initial VAE model on $\boldsymbol{\tau}$ (Alg. A1)
6: Identify the compact representations $\boldsymbol{s}^{min}$ and change factors $\boldsymbol{\theta}^{min}$ based on $\boldsymbol{C}^{\cdot \rightarrow \cdot}$
7: # Model estimation & policy learning
8: **for** $n = 0, \ldots, N - 1$ **do**
9:    **for** $t = 0, \ldots, H - 1$ **do**
10:       Observe $\boldsymbol{s}_t$ from Env;
11:       # Estimating latent change factors
12:       **if** $n \cdot (H - 1) + t \in \tilde{\boldsymbol{t}}$ **then**
13:          $m \leftarrow m + 1$
14:          **for** j = s, r **do**
15:             Infer $\mu_{\gamma^j}(\boldsymbol{\theta}_{t_{m-1}}^j)$ and $\sigma_{\gamma^j}^2(\boldsymbol{\theta}_{t_{m-1}}^j)$ via $p_{\gamma^j}$
16:             Sample $\boldsymbol{\theta}_{t_m}^j \sim \mathcal{N}\left(\mu_{\gamma^j}(\boldsymbol{\theta}_{t_{m-1}}^j), \sigma_{\gamma^j}^2(\boldsymbol{\theta}_{t_{m-1}}^j)\right)$
17:          **end for**
18:       **end if**
19:       Generate $\boldsymbol{a}_t \sim \pi_\psi(\boldsymbol{a}_t \mid \boldsymbol{s}_t^{min}, \boldsymbol{\theta}_{t_m}^{min})$
20:       Receive $r_{n,t}$ from Env
21:       Add $(\boldsymbol{s}_t, \boldsymbol{a}_t, r_t, \boldsymbol{\theta}_{t_m}^s, \boldsymbol{\theta}_{t_m}^r)$ to replay buffer $\mathcal{D}$;
22:       Extract a trajectory with length $k$ from $\mathcal{D}$;
23:       Learn VAE (Alg. A1) with updateG=False;
24:       Sample a batch of data from $\mathcal{D}$
25:       Update policy network parameters $\psi$
26:    **end for**
27: **end for**

---

only infer $\theta$ using via CF dynamics networks at change points. Furthermore, we also adjust the objective functions of FN-VAE to fit the discrete changes. At timestep $t$ in episode $n$, where $t_m \leq \big((n-1) \cdot H + t\big) < t_{m+1}$, we have:

• Prediction and reconstruction losses:

$$
\begin{aligned}
\mathcal{L}_{\text{rec-dyn}} &= \sum_{t=1}^{T-2} \mathbb{E}_{\theta_{t_m}^s \sim q_\phi} \log p_{\alpha_1}(\boldsymbol{s}_t | \boldsymbol{s}_{t-1}, \boldsymbol{a}_{t-1}, \boldsymbol{\theta}_{t_m}^s; \boldsymbol{C}^{\rightarrow s}) \\
\mathcal{L}_{\text{pred-dyn}} &= \sum_{t=1}^{T-2} \mathbb{E}_{\theta_{t_m}^s \sim q_\phi} \log p_{\alpha_2}(\boldsymbol{s}_{t+1} | \boldsymbol{s}_t, \boldsymbol{a}_t, \boldsymbol{\theta}_{t_m}^s)
\end{aligned}
\tag{A3}
$$

$$
\begin{aligned}
\mathcal{L}_{\text{rec-rw}} &= \sum_{t=1}^{T-2} \mathbb{E}_{\theta_{t_m}^r \sim q_\phi} \log p_{\beta_1}(r_t | \boldsymbol{s}_t, \boldsymbol{a}_t, \boldsymbol{\theta}_{t_m}^r; \boldsymbol{c}^{s \rightarrow r}, \boldsymbol{c}^{a \rightarrow r}) \\
\mathcal{L}_{\text{pred-rw}} &= \sum_{t=1}^{T-2} \mathbb{E}_{\theta_{t_m}^r \sim q_\phi} \log p_{\beta_2}(r_{t+1} | \boldsymbol{s}_{t+1}, \boldsymbol{a}_{t+1}, \boldsymbol{\theta}_{t_m}^r)
\end{aligned}
\tag{A4}
$$

• KL loss:

$$
\begin{aligned}
\mathcal{L}_{\text{KL}} = \sum_{t=2}^{T} &\text{KL}\big(q_{\phi^s}(\boldsymbol{\theta}_{t_m}^s | \boldsymbol{\theta}_{t_{m-1}}^s, \boldsymbol{\tau}_{0:t})) \| p_{\gamma^s}(\boldsymbol{\theta}_{t_m}^s | \boldsymbol{\theta}_{t_{m-1}}^s; \boldsymbol{C}^{\theta^s \rightarrow \theta^s})) \\
&+ \text{KL}\big(q_{\phi^r}(\boldsymbol{\theta}_{t_m}^r | \boldsymbol{\theta}_{t_{m-1}}^r, \boldsymbol{\tau}_{0:t})) \| p_{\gamma^r}(\boldsymbol{\theta}_{t_m}^r | \boldsymbol{\theta}_{t_{m-1}}^r; \boldsymbol{C}^{\theta^r \rightarrow \theta^r}))
\end{aligned}
\tag{A5}
$$

• Sparsity loss:

$$\mathcal{L}_{\text{sparse}} = w_1 \|\boldsymbol{C}^{\boldsymbol{s} \to \boldsymbol{s}}\|_1 + w_2 \|\boldsymbol{C}^{\boldsymbol{a} \to \boldsymbol{s}}\|_1 + w_3 \|\boldsymbol{C}^{\boldsymbol{\theta}^{\boldsymbol{s}} \to \boldsymbol{s}}\|_1$$
$$+ w_6 \|\boldsymbol{C}^{\boldsymbol{\theta}^{\boldsymbol{s}} \to \boldsymbol{\theta}^{\boldsymbol{s}}}\|_1 + w_7 \|\boldsymbol{C}^{\boldsymbol{\theta}^{\boldsymbol{r}} \to \boldsymbol{\theta}^{\boldsymbol{r}}}\|_1 \qquad \text{(A6)}$$
$$+ w_4 \|\boldsymbol{c}^{\boldsymbol{s} \to \boldsymbol{r}}\|_1 + w_5 \|\boldsymbol{c}^{\boldsymbol{a} \to \boldsymbol{r}}\|_1$$

• Smoothness loss:

$$\mathcal{L}_{\text{smooth}} = \sum_{t=2}^{T} \left( \|\boldsymbol{\theta}^{\boldsymbol{s}}_{t_m} - \boldsymbol{\theta}^{\boldsymbol{s}}_{t_{m-1}}\|_1 + \|\boldsymbol{\theta}^{\boldsymbol{r}}_{t_m} - \boldsymbol{\theta}^{\boldsymbol{r}}_{t_{m-1}}\|_1 \right) \qquad \text{(A7)}$$

The total loss $\mathcal{L}_{\text{vae}} = k_1 \mathcal{L}_{\text{rec}} + k_2 \mathcal{L}_{\text{pred}} - k_3 \mathcal{L}_{\text{KL}} - k_4 \mathcal{L}_{\text{sparse}} - k_5 \mathcal{L}_{\text{smooth}}$, where $k_1$, $k_2$, $k_3$, $k_4$, and $k_5$ are adjustable hyper-parameters to balance the objective functions.

### D.3 The framework dealing with raw pixels

We augment the generative process in Eq. 1-3 with the generative process of observation.

$$o_t = u_i(c_i^{\boldsymbol{s} \to \boldsymbol{o}} \odot \boldsymbol{s}_t, \epsilon_t^o), \qquad \text{(A8)}$$

where $u$ is a non-linear function and $i = 1, \ldots, d$. $\boldsymbol{c}^{\boldsymbol{s} \to \boldsymbol{o}} := [c_i^{\boldsymbol{s} \to \boldsymbol{o}}]_{i=1}^d$. $\epsilon_t^o$ is an i.i.d. random noise. To learn the $u_i$, we model the states as the latent variables in FN-VAE. Fig. A10 gives the modified

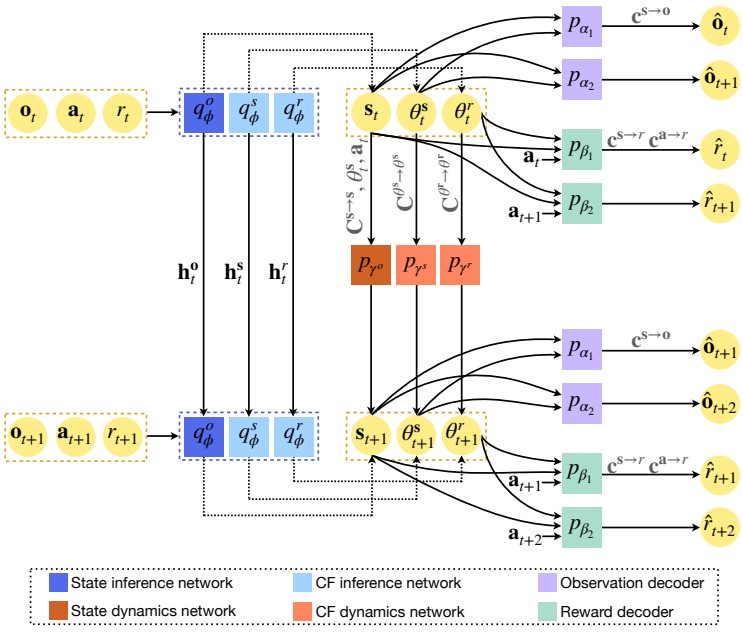

Figure A10: The architecture of FN-VAE using raw pixel as input.

FN-VAE dealing with raw pixels, where the states are also in the latent space. Different from the original FN-VAE, we incorporate state inference networks and state dynamics networks. Moreover, we reconstruct and predict the current and future observations using the observation decoder. Detailed objective functions are given below.[1]

• Prediction and reconstruction losses

---

[1]Here we give the example of handling the discrete changes.

$$\mathcal{L}_{\text{rec-obs}} = \sum_{t=1}^{T-2} \mathbb{E}_{s_t \sim q_{\phi^o}} \log p_{\alpha_1}(\boldsymbol{o}_t | \boldsymbol{s}_t; \boldsymbol{c}^{\boldsymbol{s} \to \boldsymbol{o}})$$

$$\mathcal{L}_{\text{pred-obs}} = \sum_{t=1}^{T-2} \mathbb{E}_{s_t \sim q_{\phi^o}} \log p_{\alpha_2}(\boldsymbol{o}_{t+1} | \boldsymbol{s}_t, \boldsymbol{\theta}_{t_m}^{\boldsymbol{s}}) \tag{A9}$$

$$\mathcal{L}_{\text{rec-rw}} = \sum_{t=1}^{T-2} \mathbb{E}_{(\theta_{t_m}^r \sim q_\phi, s_t \sim q_{\phi^o})} \log p_{\beta_1}(r_t | \boldsymbol{s}_t, \boldsymbol{a}_t, \boldsymbol{\theta}_{t_m}^r; \boldsymbol{c}^{\boldsymbol{s} \to r}, \boldsymbol{c}^{\boldsymbol{a} \to r})$$

$$\mathcal{L}_{\text{pred-rw}} = \sum_{t=1}^{T-2} \mathbb{E}_{(\theta_{t_m}^r \sim q_\phi, s_t \sim q_{\phi^o})} \log p_{\beta_2}(r_{t+1} | \boldsymbol{s}_t, \boldsymbol{a}_{t+1}, \boldsymbol{\theta}_{t_m}^r) \tag{A10}$$

• KL loss

$$\mathcal{L}_{\text{KL}} = \sum_{t=2}^{T} \text{KL}\big(q_{\phi^s}(\boldsymbol{\theta}_{t_m}^{\boldsymbol{s}} | \boldsymbol{\theta}_{t_{m-1}}^{\boldsymbol{s}}, \boldsymbol{\tau}_{0:t})) \| p_{\gamma^s}(\boldsymbol{\theta}_{t_m}^{\boldsymbol{s}} | \boldsymbol{\theta}_{t_{m-1}}^{\boldsymbol{s}}; \boldsymbol{C}^{\boldsymbol{\theta}^{\boldsymbol{s}} \to \boldsymbol{\theta}^{\boldsymbol{s}}})\big)$$

$$+ \text{KL}\big(q_{\phi^r}(\boldsymbol{\theta}_{t_m}^r | \boldsymbol{\theta}_{t_{m-1}}^r, \boldsymbol{\tau}_{0:t})) \| p_{\gamma^r}(\boldsymbol{\theta}_{t_m}^r | \boldsymbol{C}^{\boldsymbol{o}_t \to \boldsymbol{\theta}^r})\big) \tag{A11}$$

$$+ \text{KL}\big(q_{\phi^o}(\boldsymbol{s}_t | \boldsymbol{\tau}_{0:t}, \boldsymbol{\theta}_{t_m}^{\boldsymbol{s}})) \| p_{\gamma^o}(\boldsymbol{s}_t | \boldsymbol{s}_{t-1}, \boldsymbol{a}_{t-1}, \boldsymbol{\theta}_{t_m}^{\boldsymbol{s}}; \boldsymbol{C}^{\boldsymbol{s} \to \boldsymbol{s}}, \boldsymbol{C}^{\boldsymbol{a} \to \boldsymbol{s}}, \boldsymbol{C}^{\boldsymbol{\theta}^{\boldsymbol{s}} \to \boldsymbol{s}})\big)$$

where $\boldsymbol{\tau}_{0:t} = \{\boldsymbol{o}_0, r_0, \boldsymbol{o}_1, r_1, \ldots, \boldsymbol{o}_t, r_t\}$.

• Sparsity loss

$$\mathcal{L}_{\text{sparse}} = w_1 \|\boldsymbol{C}^{\boldsymbol{s} \to \boldsymbol{s}}\|_1 + w_2 \|\boldsymbol{C}^{\boldsymbol{a} \to \boldsymbol{s}}\|_1 + w_3 \|\boldsymbol{C}^{\boldsymbol{\theta}^{\boldsymbol{s}} \to \boldsymbol{s}}\|_1$$

$$+ w_4 \|\boldsymbol{c}^{\boldsymbol{s} \to r}\|_1 + w_5 \|\boldsymbol{c}^{\boldsymbol{a} \to r}\|_1 \tag{A12}$$

$$+ w_6 \|\boldsymbol{C}^{\boldsymbol{\theta}^{\boldsymbol{s}} \to \boldsymbol{\theta}^{\boldsymbol{s}}}\|_1 + w_7 \|\boldsymbol{C}^{\boldsymbol{\theta}^r \to \boldsymbol{\theta}^r}\|_1 + w_8 \|\boldsymbol{C}^{\boldsymbol{s} \to \boldsymbol{o}}\|_1$$

• Smooth loss

$$\mathcal{L}_{\text{smooth}} = \sum_{t=2}^{T} \Big( ||\boldsymbol{\theta}_{t_m}^{\boldsymbol{s}} - \boldsymbol{\theta}_{t_{m-1}}^{\boldsymbol{s}}||_1 + ||\boldsymbol{\theta}_{t_m}^r - \boldsymbol{\theta}_{t_{m-1}}^r||_1 \Big) \tag{A13}$$

### D.4 Hyper-parameter selection

#### D.4.1 Factored model estimation

**Input with symbolic states.** In Half-Cheetah, Sawyer-Reaching, and Minitaur the symbolic states are observable. For the CF dynamic networks, we use 2-layer fully connected networks. The number of neurons is $512$. For CF inference networks, we use 2-layer fully connected networks, where the number of neurons is $256$, followed by the LSTM networks with $256$ hidden units. The initial learning rates for all losses are set to be $0.1$ with a decay rate $0.99$. The batch size is $256$ and the length of time steps is equal to the horizon in each task. The number of RNN cells is $256$. The decoder networks are 2-layer fully connected networks. The number of neurons is $512$.

**Input with raw pixels.** In Saywer-Peg, we directly learn and adapt in non-stationary environments with raw pixels observed. Different from other experiments, we use the architecture described in Fig. A10. At timestep $t$, we stack 4 frames as the input $\boldsymbol{o}_t$. A 5-layer convolutional networks is used to extract the features of the trajectories of observations and rewards. The layers have 32, 64, 128, 256, and 256 filters. And the corresponding filter sizes are 5, 3, 3, 3, 4. The observation decoders are the transpose of the convolutional networks. Then the extracted features are used as the input of LSTM networks in state inference networks. The state inference networks and state dynamic networks share the same architectures with the CF inference and dynamics networks, respectively. We use the same CF inference networks, CF dynamics networks, and reward decoders with those in cases with symbolic states as input. The number of latent features is 40.

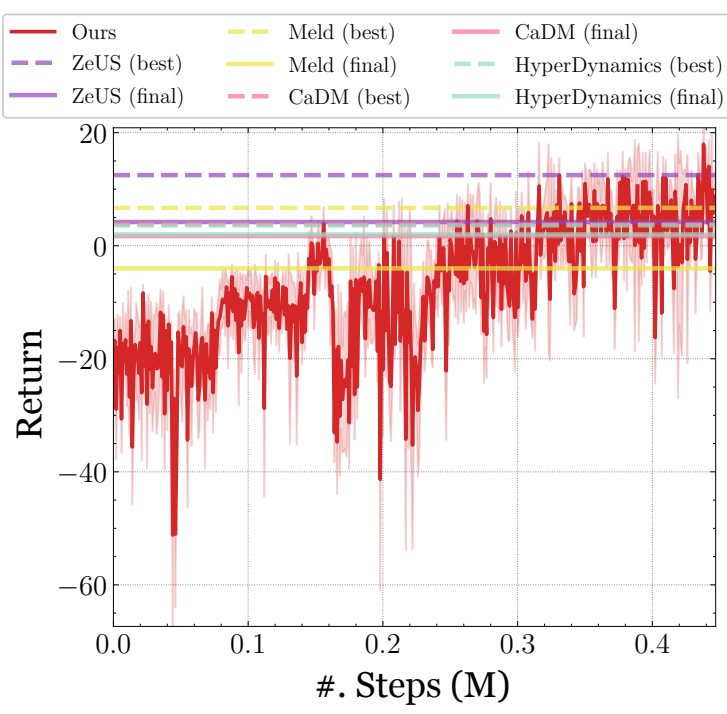

Figure A11: Average return across 10 runs on Sawyer-Peg (raw pixels) with across-episode changes.

**Balancing parameters in losses** For all experiments:

- All $w.$ are set to be $0.1$;
- Weights of the reconstruction loss: $k_1 = 0.8$;
- Weights of the prediction loss: $k_2 = 0.8$;
- Weights of KL loss: $k_3 = 0.5$;
- Weights of sparsity loss: $k_4 = 0.1$;
- Weights of smooth loss: $k_5 = 0.02$.

We use the automatic weighting method in [16] to learn the weights for $k_1, \ldots, K_5$ and grid search for $w_1, \ldots, w_7$.

**Model initialization.** Table A4, A5, and A6 provide the settings of learning the model initialization.

| | CONT (D) | A-EP (D) | W-EP (D) | A-EP (R) | A-EP (R+D) |
|---|---|---|---|---|---|
| # trajectories | 500 | 20 | 20 | 20 | 100 |
| # steps in each episode | 50 | 50 | 50 | 50 | 50 |
| # episodes | 10 | 100 | 100 | 100 | 100 |

Table A4: The selected hyper-parameters for model estimation in Half-Cheetah experiment.

| | Sawyer-Reaching | Sawyer-Peg |
|---|---|---|
| # trajectories | 500 | 20 |
| # steps in each episode | 150 | 40 |
| # episodes | 10 | 100 |

Table A5: The selected hyper-parameters for model estimation in Saywer experiments.

|  | CONT (D) | W-EP (D) | A-EP (R+D) |
|---|---|---|---|
| # trajectories | 500 | 50 | 80 |
| # steps in each episode | 100 | 100 | 100 |
| # episodes | 10 | 50 | 100 |

Table A6: The selected hyper-parameters for model estimation in Minitaur experiments.

### D.4.2 Policy learning

In the Half-Cheetah, Sawyer-Reaching, and Minitaur experiments, we follow the learning rates selection for policy networks in [7]. In Sawyer-Peg, for both actor and critic networks, we use 2-layer fully-connected networks. The number of neurons is $256$. For all experiments, we use standard Gaussian to initialize the parameters of policy networks. The learning rate is $3e - 4$. The relay buffer capacity is $50, 000$. The number of batch size is $256$.

**Details on TRIO and VariBAD.** For TRIO and VariBAD, we meta-train the models (batch size: 5000, # epochs: 2 for all experiments) and show the learning curves of meta-testing. The tasks parameters for meta-training are uniformly sampled from a Gaussian distribution. For all approaches, we use the same set of hyper-parameters for policy optimization modules (i.e., SAC). For the latent parameters in TRIO, we follow the original paper where the latent space from the inference network is projected to a higher dimension. The number of latent parameters for TRIO is the same as those in other approaches (Half-Cheetah and Minitaur: $40$, Sawyer-Reaching: $20$). We compared with TS-TRIO, with the kernels set as in the original implementation.

## E  Experimental Platforms and Licenses

### E.1  Platforms

All methods are implemented on 8 Intel Xeon Gold 5220R and 4 NVidia V100 GPUs.

### E.2  Licenses

In our code, we have used the following libraries which are covered by the corresponding licenses:

- Tensorflow (Apache License 2.0),
- Pytorch (BSD 3-Clause "New" or "Revised" License),
- OpenAI Gym (MIT License),
- OpenCV (Apache 2 License),
- Numpy (BSD 3-Clause "New" or "Revised" License)
- Keras (Apache License).