# OpenReview forum: "Factored Adaptation for Non-Stationary Reinforcement Learning"
_NeurIPS.cc/2022/Conference — NeurIPS 2022 Accept_

### Official Review · Reviewer_3UJG · 2022-07-10

**Rating:** 7
**Confidence:** 4
**Soundness:** 3 good
**Presentation:** 3 good
**Contribution:** 3 good

**Summary:**

This paper proposes a novel formalism to model non-stationary environments, Factored Non-stationary Markov Decision Process (FN-MDP), that models latent factors that affect the dynamics and rewards and evolves with time as in Dynamic Bayesian Network. Then, they propose FN-VAE, which is built on top of AdaRL, and is used to learn the parameters (latent factors, dynamics, rewards, masks) of the underlying FN-MDP. The method is evaluated in several robotic environments and compared with state-of-the-art algorithms tailored to deal with non-stationarity in RL.

**Questions:**

Furthermore, I have the following questions and constructive criticisms:

- In Definition 1, I suggest clarifying that it is assumed that the action is a vector of m dimensions. It is ambiguous as m could denote the number of discrete actions.

- “Although the masks and noise are stationary, we allow the change of graph structure and noise distributions, whose changes are captured by $\theta$ instead.”
If the masks are stationary, does it mean that if one of its values is 0 (denoting that one state variable does not influence some other state variable), then it will never have an effect, even if the dynamics change? In other words, to be able to capture all possible dynamics changes, would we need to have all masks with none zero elements?

- The proof of Proposition 1 in the Appendix ends with: “In this setting, identifiability of the graph G is trivial [6].” However, It is not clear why this is trivial. As this is an important theoretical result of the paper, I suggest elaborating this proof.

- The role of the state and reward encoders in Section 3 is not very clear. Why a single prediction network, conditioned on the masks, would be not enough to predict the next state? How are the encoders used?

- How can we obtain the adjustable parameters $w_1$ … $w_7$? In the total loss, how can we obtain $k_1$ … $k_5$?

- The total objective function in line 167 does not include $\mathcal{L}_{\text{rec-dyn}}$. Additionally, there is a typo here: the coefficient $k_5$ does not appear in the loss.

- “the policy parameters = $\psi = (\pi, Q)$ are the actor $L_\pi$ and critic loss $L_Q$.” I believe this sentence is not well formulated, how can a loss be a parameter?

- In Algorithm 1, how is $s^{min}_t$ and $\theta^{min}_t$ computed before following the policy in line 23?

- “The horizon in each episode is 50.” The horizon in these environments is generally higher, why this value was chosen?

- In Fig. 2(g-j) how were the values normalized? I suppose the value of 1.0 does not mean that the proposed method reaches the maximum possible reward, but that it is the maximum reference point used in the normalization.

- Because of the causality assumptions (see Section A of the Appendix), the proposed model can not model different types of non-stationarity at the same time (e.g. wind and gravity). Are any of the related work (e.g. LILAC) able to deal with these changes? If so, perhaps an experiment in such a setting would be interesting to observe whether the proposed method could still outperform the other baselines.

- For completeness of the first sentence of the Related Work section, the paper [“Minimum-Delay Adaptation in Non-Stationary Reinforcement Learning via Online High-Confidence Change-Point Detection.” In Proceedings of the 20th International Conference on Autonomous Agents and MultiAgent Systems] is a more recent work on non-stationarity that detects changes that have already happened.


**Limitations:**

The authors appropriately discuss the limitations of the proposed approach. I would suggest, however, that the authors briefly include in the main text some of the assumptions of the method which are only discussed in Appendix A, as I believe they are very relevant and readers often do not check the Appendix.

**Strengths And Weaknesses:**

Strengths:
- The proposed formalism is able to model, in a factored manner, non-stationary environments in which the factors that change the dynamics/reward also evolve over time. This is a novel idea not yet explored in the related literature.
- The authors compare the proposed approach with several state-of-the-art algorithms tailored to deal with non-stationarity, showing relevant performance improvements. Several ablation experiments were also presented.

Weaknesses:
- Some mathematical and algorithmic definitions are not very clear in the current version and should be clarified in the main text.

---

> ### Author Response · Authors · 2022-07-30
> **Response to Reviewer 3UJG (1/3)**
>
> Thanks for your thorough review. We tried to clarify the definitions and theory (highlighted in blue in the revised version) and fixed the typos. We also would like to point out that the assumptions we discuss in the Broader impacts are already mentioned in the main paper. We have added a few clarifications in the Broader impacts, but just for completeness:
> - no unobserved confounders (except for the change factors) and no instantaneous causal effects between the state components are implied by Definition 1, since the FN-MDP has a Dynamic Bayesian Network $\mathcal{G}$ in which only the change factors are unobserved.
> This is also stated in a equivalent form in the equations of the generative model, e.g. in Eq. (1) in which the state component $s_{i,t}$ can only depend on components of the state at the previous time-step $\mathbf{s}\_{t-1}$;
> - the causal Markov and faithfulness assumptions are used for Proposition 1 and 2 to prove that we can recover (most of the) true causal graph.
>
> > Q1: In Definition 1, I suggest clarifying that it is assumed that the action is a vector of m dimensions. It is ambiguous as m could denote the number of discrete actions.
>
> Thanks for pointing this out. We have clarified this in the revised version.
>
> > Q2: "Although the masks and noise are stationary, we allow the change of graph structure and noise distributions, whose changes are captured by $\theta$ instead." If the masks are stationary, does it mean that if one of its values is 0 (denoting that one state variable does not influence some other state variable), then it will never have an effect, even if the dynamics change? In other words, to be able to capture all possible dynamics changes, would we need to have all masks with none zero elements?
>
> Thanks for pointing this out. We clarified this part in the revised version in Section 2 and rephrased parts of the Broader Impacts in the Appendix that might be confusing. For the text in Section 2 we added:
> "Although $\mathbf{c}^{\cdot \rightarrow \cdot}$ and $\epsilon$ are stationary, we model the changes in the functions and some changes in the graph structure through $\mathbf{\theta}$.
> For example a certain value of $\mathbf{\theta}\_t^r$ can switch off the contribution of some of the state or action dimensions in the reward function, or in other words nullify the effect of some edges in $\mathcal{G}$. Similarly the contribution of the noise distribution to each function can be modulated via the change factors. On the other hand, this setup does not allow adding edges that are not captured by the binary masks $\mathbf{c}^{\cdot \rightarrow \cdot}$."
> In particular, in our current method, the change in dynamics can only switch off edges from the estimated mask, but not switch them on. On the other hand, the estimated mask will contain the union of the edges that are present at any timestep that is used for model estimation. We assume this is a sensible inductive bias in setting. One could estimate also the graph dynamically (e.g. by allowing it to change at each iteration in Alg.1), but might would also require changing the compact representation $\mathbf{s}\_{t}^{\min }$ and $\mathbf{\theta}\_{t}^{\min }$ (which are estimated based on the graph), and therefore make policy optimization harder.
> If we want to capture all possible dynamic changes, including the ones that we might not observe, we would then indeed need binary masks without any zeros. In our setting, this is represented by the ablation that does not use the structure.
>
> >Q3: The proof of Proposition 1 in the Appendix ends with: "In this setting, identifiability of the graph G is trivial [6]." However, It is not clear why this is trivial. As this is an important theoretical result of the paper, I suggest elaborating this proof.
>
> Thanks for pointing this out, we have completely reworked the proof and made several of the points much more explicit. We do not claim that this proof is in itself a particularly novel result, since the identifiability of similar time-series with no unobserved confounders and no instantaneous effects is well-known, but adding the change factors which do have some instantaneous effects on the state and reward did require a bit of additional explanation.

---

> > ### Author Response · Authors · 2022-07-30
> > **Response to Reviewer 3UJG (2/3)**
> >
> > > Q4: The role of the state and reward encoders in Section 3 is not very clear. Why a single prediction network, conditioned on the masks, would be not enough to predict the next state? How are the encoders used?
> >
> > We use the separate state and reward encoders to disentangle the data generation process of dynamics and rewards in FN-VAE. To verify the effectiveness of the disentangling design, we conducted an ablation on using mixed latent features and only one encoders and single reconstruction/prediction network in the model. We ran the experiments on the scenarios with multiple change factors on both dynamics and rewards in Minitaur benchmark. The results below suggests that the disentangling design can bring more performance gain.
> >
> >
> > |       Methods      | Average final rewards |
> > |:------------------:|:------------------------:|
> > | Mixed latent space |      26.9 (+/-12.6)      |
> > |        Ours        |       40.2 (+/-5.3)      |
> >
> > >Q5: How can we obtain the adjustable parameters $w_{1} \ldots w_{7}$ ? In the total loss, how can we obtain $k_{1} \ldots k_{5}$ ?
> >
> > We have added an explanation in the revised version of the paper. In general, we use the automatic weighting method in [Liebel et al 2018] to learn the weights for $k_1, \ldots, k_7$ and grid search for $w_1, \ldots, w_7$. We have also clarified it in more detail in the updated Supplementary Section D.4.1.
> >
> > [Liebel et al 2018] Liebel, Lukas, and Marco Körner. "Auxiliary tasks in multi-task learning." arXiv preprint arXiv:1805.06334 (2018).
> >
> > >Q6: The total objective function in line 167 does not include $\mathcal{L}\_{\text {rec-dyn}}$. Additionally, there is a typo here: the coefficient $k\_{5}$ does not appear in the loss.
> >
> > Thanks for pointing this out. We used the same weights for $\mathcal{L}\_{\text {rec-dyn}}$ and $\mathcal{L}\_{\text {rec-rw }}$. The total loss is $\mathcal{L}\_{\text {vae }}=k\_{1}\left(\mathcal{L}\_{\text {rec-dyn}}+\mathcal{L}\_{\text {rec-rw}}\right)+k\_{2}\left(\mathcal{L}\_{\text {pred-dyn }}+\mathcal{L}\_{\text {pred-rw }}\right)-k\_{3} \mathcal{L}\_{\text {KL }}-k\_{4} \mathcal{L}\_{\text {sparse }}-k\_{5} \mathcal{L}\_{\text {smooth }}$.
> > We have clarified this and fixed the typo in the revised version.
> >
> > >Q7: "the policy parameters $\psi=(\pi, Q)$ are the actor $L_{\pi}$ and critic loss $L_{Q}$." I believe this sentence is not well formulated, how can a loss be a parameter?
> >
> > Sorry for the ambiguity. We have corrected the typo in the revised version.
> >
> > >Q8: In Algorithm 1, how is $\mathbf{s}\_{t}^{\min }$ and $\mathbf{\theta}\_{t}^{\min }$ computed before following the policy in line 23?
> >
> > $\mathbf{s}^{\min }$ and $\mathbf{\theta}^{\min }$ are the dimensions of the state and change factors that are minimal and sufficient for policy optimization, as shown in AdaRL [Huang et al. 2022]. They can be identified from the estimated binary masks/graph, by selecting all variables that have a directed path to a present or future reward. For completeness, we have added an explanation in Section 2.
> >
> > In particular, in Algorithm 1, $\mathbf{s}^{\min }$ and $\mathbf{\theta}^{\min }$ are computed in Line 5, after we have estimated the binary masks/the graph. In Line 23 we select the values at time t of these dimensions, $\mathbf{s}\_{t}^{\min }$ and $\mathbf{\theta}\_{t}^{\min }$.
> >
> > >Q9: "The horizon in each episode is 50." The horizon in these environments is generally higher, why this value was chosen?
> >
> > We chose this setting to follow the instructions in LILAC [Xie et al., 2021] for a fair comparison, where each episode is 50 time-steps long.
> >
> > [Xie et al., 2021] Xie, Annie, James Harrison, and Chelsea Finn. "Deep reinforcement learning amidst lifelong non-stationarity." ICML 2021.
> >
> > >Q10: In Fig. 2(g-j) how were the values normalized? I suppose the value of $1.0$ does not mean that the proposed method reaches the maximum possible reward, but that it is the maximum reference point used in the normalization.
> >
> > That's correct, the values are normalized based on the results of our method. We chose this normalization, because our approach can outperform all the baseline methods (e.g., LILAC, TRIO, and VariBAD) and it seems an intuitive measure. For completeness, we added the figures where all methods are normalized based on the oracle in Appendix Fig.A7-A8.

---

> > > ### Author Response · Authors · 2022-07-30
> > > **Response to Reviewer 3UJG (3/3)**
> > >
> > > >Q11: Because of the causality assumptions (see Section A of the Appendix), the proposed model can not model different types of non-stationarity at the same time (e.g. wind and gravity). Are any of the related work (e.g. LILAC) able to deal with these changes? If so, perhaps an experiment in such a setting would be interesting to observe whether the proposed method could still outperform the other baselines.
> > >
> > > Our model, LILAC, TRIO and VariBad can model multiple concurrent changes. We show the experiments with multiple change factors in Fig.2 (j). We describe the results in the paragraph "Multiple change factors" at the end of the Evaluation section. In particular, we test "1) only change wind forces (1D); 2) change wind forces and gravity concurrently (2D); 3) change wind force and target speed (1D+1R); and 4) change wind force, gravity, and target speed together (2D+1R) in an across-episode way in Half-Cheetah."
> > > In the multiple changes setting, FANS-RL outperforms the baselines even more than in the single change setting.
> > > We also rephrased a few sentences in the Broader Impact section of the Appendix that might have been confusing from this perspective. In short, we can model multiple concurrent changes, but in our current method we cannot model changes that introduce new edges in the graph $\mathcal{G}$ that we haven't observed at model estimation time.
> > >
> > > >Q12: For completeness of the first sentence of the Related Work section, the paper ["Minimum-Delay Adaptation in Non-Stationary Reinforcement Learning via Online High-Confidence Change-Point Detection." In Proceedings of the 20th International Conference on Autonomous Agents and MultiAgent Systems] is a more recent work on non-stationarity that detects changes that have already happened.
> > >
> > > Thanks for the pointer. We discussed it in the updated related work section.

---

> > > > ### Comment · Reviewer_3UJG · 2022-08-05
> > > > **Response to Authors**
> > > >
> > > > I thank the authors for carefully addressing my questions and suggestions. Because I do not have further concerns, I will increase my score.

---

> > > > > ### Author Response · Authors · 2022-08-09
> > > > > **Thanks for your response**
> > > > >
> > > > > We would like to express our sincere thanks for your positive feedback and valuable suggestions.
> > > > > - As you suggested in Q4, we have updated a new revision, which includes the ablation studies on the disentangled design of CF inference networks for all scenarios (see updated Fig. 2(d) and Sec. C.5).
> > > > > - Based on your advice, we plan to include the key assumptions of the method in Appendix A in our final main text if accepted (as NeurIPS allows one additional page for the camera-ready version).

---

### Official Review · Reviewer_siKe · 2022-07-11

**Rating:** 6
**Confidence:** 3
**Ethics Flag:** Yes
**Soundness:** 3 good
**Presentation:** 2 fair
**Contribution:** 3 good

**Summary:**

In this paper, the authors considered non-stationarity which is faced when RL is deployed into the real world. To model the non-stationarity across episodes or within an episode, which seems to be general scenarios, the authors disentangled the non-stationarity as two types of latent change factors and formalized FN-MDPs. These disentangled latent factors enable the proposed method called FANS-RL to estimate changes both in the environment dynamics and the reward function. The authors assumed the generative process of FN-MDPs containing these latent factors, and FANS-RL learns the generative process of FN-MDPs during training. FANS-RL can also infer the change factors in environment dynamics and reward function at the current time-step via the learned generative process of FN-MDPs. The experimental results showed that FANS-RL outperforms baseline algorithms in various simulation tasks. Especially, the ablation study showed the performance effect of their method, which has many loss functions and variables they considered.



**Questions:**

FANS-RL can address non-stationarity, which is an important challenge in RL, and I think it is a great approach. However, it seems to rely on AdaRL, so I need additional explanation and experimental results to support the acceptance of the paper.

1. Why did the authors formalize FN-MDPs, not N-MDPs? Although the ablation study without structure shows improvement with respect to the expected return, I need a brief and intuitive explanation for understanding the effect of factored MDP.

2. As mentioned in “Weakness”,  the experimental results showed the adaptation to the limited changes in the environment dynamics. To find out the adaptation for the changes in the agent’s mechanism, I suggest additional experiments for these changes; for example, the increasing joint friction is introduced at a joint of Half-Cheetah.

3. To check the effect of disentangling the latent change factor as the two ones, I suggest adding an ablation study without disentangling latent factors in figure 2-(d).

4. Figure 2-(e,f) shows the distances of the learned $\theta^r$, similarly, I suggest plotting the distances of the learned $\theta^s$; for example, the $\theta^s$ estimates the changing wind force. It is helpful to identify that the learned latent factor can capture the true change in environment dynamics.

5. In Figure 2-(g), is the number of latent features of FANS-RL total number of dimensions of two latent features $\theta^s$ and $\theta^r$?

6. What is the fundamental difference between the FN-VAE and the MiSS-VAE of AdaRL?



**Ethics Review Area:**

["Research Integrity Issues (e.g., plagiarism)"]

**Limitations:**

I already mentioned the limitations of this paper and the suggestions for the proposed method.

**Strengths And Weaknesses:**

This paper proposed FANS-RL that addresses non-stationarity both in environment dynamics and reward function.

1. Strengths

(1-1) The authors formalized FN-MDPs that can deal with a general non-stationary RL setting including changes both across episodes and within an episode. It is a very important problem in RL.
(1-2) FN-MDPs also contain two latent change factors for the environment dynamics and the reward function, and FANS-RL can more explicitly handle the non-stationarity of the MDP due to the two change factors.
(1-3) FANS-RL shows better adaptation for non-stationarity than other existing algorithms in various tasks. Also, the ablation study showed the effect of each component of FANS-RL architecture with respect to the expected return.

2. Weakness

(2-1) The proposed method showed the adaptation for the limited changes in the environment dynamics; for example, the changes of wind force or agent’s mass or gravity. So, the reviewer wonders if FANS-RL can adapt to changes in the agent’s mechanism; for example, the disabled joint of the agent due to aging.

(2-2) It seems difficult to find the fundamental difference between the architecture of FN-VAE and that of MiSS-VAE of AdaRL[1], even though they learn different MDPs. They have too similar components and loss functions.

(2-3) The paper relies much on the definitions and theorems of AdaRL [1] in literature. It would be better to add an additional explanation of the shared representation $s^{min}$ and the compact domain-specific representation $\theta^{min}$.
Especially, in Line 74-84 in Appendix B.2, there exist sentences verbatim to those in  the proof of theorem1, page 18, [1].  I marked ethics alert  due to this point. Please check if this is OK.
The paragraph of concern: "We denote the variable set in the system ...."

3. minor points (typo)
(3-1) The first equation between line 72-73: $pa(\theta_{ij,t}^s)\to pa(\theta_{j,t}^s)$
(3-2) Figure 2-(f) in the main paper: the x and y labels need to correct.

[1] Biwei Huang et al., AdaRL: What, Where, and How to Adapt in Transfer Reinforcement Learning, ICLR 2022

---

> ### Author Response · Authors · 2022-07-30
> **Response to Reviewer siKe (1/2)**
>
> Thank you for your time and attention in giving such a thoughtful review. We have made some changes to the revised version (highlighted in blue). We added an introduction on AdaRL and MiSS-VAE in the preliminary section (B.2) of the updated Appendix, focusing on the MDP case. We have also extensively reworked our proofs, and clarified how are they related to previous work. We address your specific comments one by one below.
>
> >Q1: The proposed method showed the adaptation for the limited changes in the environment dynamics; for example, the changes of wind force or agent’s mass or gravity. So, the reviewer wonders if FANS-RL can adapt to changes in the agent’s mechanism; for example, the disabled joint of the agent due to aging.
>
> Thanks for the suggestion. We are now running the experiments you recommended. We will update the results by posting a new reply once they are available.
>
> >Q2: It seems difficult to find the fundamental difference between the architecture of FN-VAE and that of MiSS-VAE of AdaRL[1], even though they learn different MDPs. They have too similar components and loss functions.
>
> Though both Miss-VAE and FN-VAE leverage factorized generative models to learn the data generation process under distribution shift in RL, several aspects are different:
>
> - As you mentioned, we are modeling different (PO)MDPs under different problem settings (transfer RL from well-defined source domains to targets versus non-stationary RL). In particular, the change factors are quite different and so they are learnt quite differently. Miss-VAE uses the domain index $k$ as the input of the model and updates $\mathbf{\theta}_k$ (which is assumed to be constant in each domain) during training. In contrast, in FN-MDPs we model the non-stationary change factors as latent variables and allow them to vary according to a Markov process. So the CF inference networks in FN-VAE use LSTMs and we also add the CF dynamics networks, that require a KL loss $\mathcal{L}\_{\text{KL}}$ that is quite different from the one in AdaRL and the smoothness loss across time-steps $\mathcal{L}\_{\text{smooth}}$;
>
> - As opposed to Miss-VAE, in FN-VAE we use separate encoders for dynamics and rewards. As we show in the answer on modelling the latent change factors of the reward and dynamics as a single multidimensional change factor, these separate encoders and decoders improve our results.
>
> - Finally, Miss-VAE is focused on pixel inputs. In the special case of transfer RL in MDPs (as opposed to FN-MDPs), there aren't any latent variables to be observed, so Miss-VAE is technically not a VAE. We also show an extension of FN-VAE to raw pixels in Appendix D.3, which is a better comparison. The prediction and reconstruction losses in that case are very similar, but to be fair they are also quite obvious/common in this setting.
>
> > Q3: The paper relies much on the definitions and theorems of AdaRL [1] in literature. It would be better to add an additional explanation of the shared representation $s^{\text {min }}$ and the compact domain-specific representation $\theta^{\text {min }}$.
> Especially, in Line 74-84 in Appendix B.2, there exist sentences verbatim to those in the proof of the the in this point. Please check if this is OK. The paragraph of concern: "We denote the variable set in the system ...."
>
> We have added a subsection to introduce the compact representations in Section 2. Regarding the proof, the part that was similar (but still with the appropriate changes related to the non-stationarity/time index vs transfer RL/ domain index) is related to the notation and the description of previous results from [Huang et al. 2020], since it's a similar setup. We have changed the proof and clarified the similarities to the related proof in AdaRL.
>
> [Huang et al. 2020] Huang, B., Zhang, K., Zhang, J., Ramsey, J. D., Sanchez-Romero, R., Glymour, C., & Schölkopf, B. (2020). Causal Discovery from Heterogeneous/Nonstationary Data. J. Mach. Learn. Res., 21(89), 1-53
>
> >Q4: minor points (typo) 1. The first equation between line 72-73: $pa\left(\theta_{i j, t}^{s}\right) \rightarrow pa\left(\theta_{j, t}^{s}\right)$ 2. Figure 2-(f) in the main paper: the $x$ and $y$ labels need to correct.
>
> Thanks for pointing this out. We have corrected them in the revised version.

---

> > ### Author Response · Authors · 2022-07-30
> > **Response to Reviewer siKe (2/2)**
> >
> > >Q5: Why did the authors formalize FN-MDPs, not N-MDPs? Although the ablation study without structure shows improvement with respect to the expected return, I need a brief and intuitive explanation for understanding the effect of factored MDP.
> >
> > We could potentially formalize N-MDPs, which would be MDPs with a latent change factor that follows a Markov process and they would be very similar to other approaches in literature, e.g. dynamic MDPs in LILAC.
> > On the other hand, our whole approach is based on factored representations and causality, following the ideas of a whole line of work Factored (PO)MDPs.
> > As shown in our experiments, learning the graphical structure is the main improvement that is derived from our framework, since it allows one to (1) capture non-stationarity in a compact way (a low-dimensional change factor), so one can adapt to it in an efficient way, and (2) identify the compact representations from AdaRL, i.e., the minimal dimensions for policy learning, thus also improving the sample efficiency.
> >
> > As an example of capturing non-stationarity in a compact way, in the robotic control task, changes on ground frictions may only affect the state variables of robots' legs while the state variables of hands or head will not be affected. Then we can learn a low-dimensional $\boldsymbol{\theta}$ connected to the state variables of legs in the graph.
> >
> > As an example of reducing the state dimensions, if the robot is trained to run at a target speed in 2D space, the position of the head may not be useful for policy learning as there is no path from the head position to the reward in the learned DBN.
> >
> > >Q6: To check the effect of disentangling the latent change factor as the two ones, I suggest adding an ablation study without disentangling latent factors in figure 2-(d).
> >
> > To verify this, we conducted an ablation study where we only have one mixed encoder and decoder for reconstruction and prediction. We ran the experiments on the scenarios with multiple change factors on both dynamics and rewards in Minitaur benchmark. We will include the full results for all settings in the final version of the paper. The results (average across $10$ runs) are given below.
> >
> > |       Methods      | Average final rewards |
> > |:------------------:|:------------------------:|
> > | Mixed latent space |      26.9 (+/-12.6)      |
> > |        Ours        |       40.2 (+/-5.3)      |
> >
> > The results verify the effectiveness of the disentangling design in our model.
> >
> > >Q7: Figure 2-(e,f) shows the distances of the learned $\theta^{r}$, similarly, I suggest plotting the distances of the learned $\theta^{s}$; for example, the $\theta^{s}$ estimates the changing wind force. It is helpful to identify that the learned latent factor can capture the true change in environment dynamics.
> >
> > Thanks for the suggestions. We verify one dynamic CF (wind forces in Half-Cheetah). We randomly sample a few data points and compute the Euclidean distance below. $D(f^{i,j}\_w)$ and $D(\theta^{s}\_{i,j})$ denotes the Euclidean distance on wind force $f_w$ and $\theta^s$ between two sampled data points. We can find that there is a positive correlation between the distance of learned $\theta^s$ and true wind forces. We add two heatmaps (similar to Fig.2(f)) in the revised Fig. A10 in the appendix.
> >
> > | $D(f^{i,j}_w)$ | $D(\theta^{s}_{i,j})$ |
> > |:--------------:|:---------------------:|
> > |      1.62      |          2.9          |
> > |      10.12     |          6.8          |
> > |      11.74     |          8.5          |
> > |      14.42     |          11.0         |
> >
> > >Q8: In Figure 2-(g), is the number of latent features of FANS-RL total number of dimensions of two latent features $\theta^{s}$ and $\theta^{r}$ ?
> >
> > Yes, the number is the total number of two latent features. We have clarified this in the text in the revised version.

---

> > > ### Author Response · Authors · 2022-08-02
> > > **Results on Q1 (changes on agent’s mechanism)**
> > >
> > > Once again, thanks for the suggestion. We have conducted experiments where one joint is randomly disabled in each episode in Half-Cheetah-v3. We compared our method with LILAC [Xie et al., 2021]. The results verify that our approach can still achieve good performance when there are changes in the agent’s mechanism. We will also compare with meta-learning approaches in the coming days and post the results once they have been done. And we plan to add this experiment to the camera-ready version if it is accepted.
> > >
> > > |        Methods       | Average final rewards (across 5 runs)|
> > > |:--------------------:|:----------------------:|
> > > | LILAC |      -19.4 (+/-11.4)     |
> > > | TRIO |      -21.9 (+/-13.0)     |
> > > | VariBAD |      -17.3 (+/-10.2)     |
> > > |  Ours  |      -15.1 (+/-9.8)     |
> > >
> > > [Xie et al., 2021] Xie, Annie, James Harrison, and Chelsea Finn. "Deep reinforcement learning amidst lifelong non-stationarity." ICML 2021.
> > >
> > > ------
> > > Updated: we have also compared the meta-learning approaches, including TRIO [Poiani et al., 2021] and VariBAD [Zintgraf et al., 2021]. During meta-training, we randomly disable the joints at random time steps. Please check the results in the updated table above.
> > >
> > > [Poiani et al., 2021] Poiani, Riccardo, Andrea Tirinzoni, and Marcello Restelli. "Meta-Reinforcement Learning by Tracking Task Non-stationarity." IJCAI 2021;
> > > [Zintgraf et al., 2021] Zintgraf, Luisa, et al. "VariBAD: Variational Bayes-Adaptive Deep RL via Meta-Learning." JMLR 2021.

---

> > > > ### Comment · Reviewer_siKe · 2022-08-09
> > > > **Response to Authors**
> > > >
> > > > I thank the authors for their response. The authors have addressed most concerns and could include the new results in their final version.

---

> > > > > ### Author Response · Authors · 2022-08-09
> > > > > **Thank you for the response**
> > > > >
> > > > > Many thanks for your response and valuable suggestions. Based on your advice in Q6, we have updated a new revision, which includes the ablation studies on the disentangled design of CF inference networks for all scenarios (see updated Fig. 2(d) and Sec. C.5).

---

### Official Review · Reviewer_T7en · 2022-07-11

**Rating:** 6
**Confidence:** 1
**Soundness:** 3 good
**Presentation:** 2 fair
**Contribution:** 3 good

**Summary:**

This paper introduces FANS-RL, a factored adaptation approach that aims to generalize to non-stationary scenarios including changes across episodes and within episodes. They formalize FN-MDPs, and prove the causal graph of the transition and reward function is identifiable. The experiments show FANS-RL outperforms the state of the art.

**Questions:**

I list a few ways in which the paper can be improved below.

- The experimental chapter devotes a lot of space to the various environmental settings, and the experimental results seem to be insufficient. More detailed discussions on the experimental results in the main paper would be more appealing.

-  The experiment section has compared different baselines with ablation experiments. However, a large number of results were presented in the Appendix section. From the main text, it's not very clear why the authors' approach is more effective than baselines, and ablation's analysis doesn't show what each component of FANS-RL does. It would be useful to extend the discussion in this section on why FANS-RL outperforms baseline methods.

- Figure 2 (c) shows that FANS-RL's reward is comparable to Oracle, even higher than Oracle. It is useful to explain why this is happening.

**Strengths And Weaknesses:**

**Originality:** Fair: The paper contributes some new ideas

**Quality:**		Good: The paper appears to be technically sound, but I have not carefully checked the details.

**Clarity:**		  Good: The paper is well organized but the analysis of the experiments' results could be improved.

**Significance:** Fair:  The paper is likely to have a moderate impact within a subfield of AI.

**Main Strengths:**

- It is a nice synthesis of causality and RL, leading to an intuitive design.
- It's fascinating to generalize a Factored-MDP to a FN-MDP, solving the non-stationary problem from a different perspective.
- If the results are robust then it could be an important and useful tool.

**Main Weakness:**

The paper is weak in a few ways. Mainly, the experiments section could include additional insight on the results as I will discuss in my question comments to the authors. As a minor point, the paper omits certain details such as a more thorough intro of AdaRL due to page limitations, which is understandable but would require more background knowledge from the readers. I suggest improving the readability in the revised version.

---

> ### Author Response · Authors · 2022-07-30
> **Response to Reviewer T7en**
>
> Thank you for your constructive comments. We have made some changes in the revised version (highlighted in blue). We also added a more thorough introduction to AdaRL in the Preliminaries section (B.2) of the revised Appendix. To simplify exposition, we only describe the MDP case, which is the only relevant for our paper.
> If the paper is accepted, we will try to integrate some parts of it, as well as some results with different non-stationary settings (Table A1 in the Appendix) and a brief explanation of the ablation studies in the additional page in the final version. We answer the questions one by one in the following:
>
> > Q1: The experimental chapter devotes a lot of space to the various environmental settings, and the experimental results seem to be insufficient. More detailed discussions on the experimental results in the main paper would be more appealing.
>
> Thanks for the suggestions. In the main paper, we include most representative results, including rewards curves (Fig.2(a-c)), ablation studies (Fig.2(d)), visualization on $\boldsymbol{\theta}$ (Fig.2(e-f)), and performances under various experimental settings (Fig.2(g-j)). Due to the limited space, we put the full results in the appendix. As NeurIPS allows an additional content page for the camera-ready version, we plan to move the quantitative results on all benchmarks with different non-stationary settings (Table A1) into the final paper.
>
> > Q2: The experiment section has compared different baselines with ablation experiments. However, a large number of results were presented in the Appendix section. From the main text, it's not very clear why the authors' approach is more effective than baselines, and ablation's analysis doesn't show what each component of FANS-RL does. It would be useful to extend the discussion in this section on why FANS-RL outperforms baseline methods.
>
> There is an ablation analysis shown in Fig.2 (d) and described in the subsection "Experimental results and ablation studies", for which the whole results are in Appendix C.5, which shows what each component of FANS-RL does.
> In particular we show what happens:
> - Without smoothness loss ($\mathcal{L}_\text{smooth}$);
> - Without structural relationships ($\mathbf{C}^{\cdot \rightarrow \cdot}$);
> - Without compact representations ($\mathbf{s}^{min}, \mathbf{\theta}^{min}$);
> - Without sparsity losses ($\mathcal{L}_{\text{sparse}}$);
> - Without reward or state prediction losses ($\mathcal{L}\_{\text{pred-rw}}$, $\mathcal{L}\_{\text{pred-dyn}}$).
>
> As shown in the ablation studies and described in the text in the main paper (which we highlighted in the revised version), the most significant gain is brought by the factored representation, which provides the structural relationships between states, actions, rewards and change factors. To make it even more clear, we plan to add the details of ablation studies (Section C.5 in Appendix) into the main paper in our final version with additional space.
>
> > Q3: Figure 2 (c) shows that FANS-RL's reward is comparable to Oracle, even higher than Oracle. It is useful to explain why this is happening.
>
> We added a clarification in the revised version in the caption of Fig. 2. To improve the readability, in this figure we only show the average of the highest rewards of Oracle across the different seeds. The quantitative results can be found in Table A1. There are some seeds in which the highest reward of FANS-RL is higher than the average highest reward for Oracle, but the average highest reward for FANS-RL (the full red line) is always lower than Oracle.

---

> > ### Author Response · Authors · 2022-08-09
> > **Thanks for the review**
> >
> > Thank you for your thoughtful review. We will be happy to discuss if you have any other concerns.

---

### Official Review · Reviewer_rs6w · 2022-07-16

**Rating:** 7
**Confidence:** 3
**Soundness:** 3 good
**Presentation:** 3 good
**Contribution:** 3 good

**Summary:**

The paper proposes a factored adaptation framework for reinforcement learning in non-stationary environments. The paper first formalizes the notion of Factored Non-stationary MDP (FN-MDP), which augments a factored MDP with time-evolving latent change factors under the Markovian assumption. The generative model in AdaRL is adapted to the time-varying setting by introducing additional equations for the latent change factors update. The causal structure is captured by a DBN, where the edges are represented using binary masks. These masks can be inferred under certain identifiability assumptions.

The paper then proposes FN-VAE to model the dynamics of the latent change factors, the state transitions, and the reward function. It includes some sparsity and smoothness regularization terms, and can be adapted for continuous or discrete changes. The FANS-RL framework combines model learning with policy optimization. The policy depends on the states and change factors for the dynamics, and the graph structure is used to select only the dimensions which affect the reward.

Experiments are performed on four popular benchmarks, which have been modified to be non-stationary with both continuous and discrete changes. The results show that FANS-RL with FN-VAE performs better than some existing methods. The paper also includes ablation studies and experiments testing various aspects of the proposed method.

**Questions:**

1. The inference networks infer $\theta_t$ from $\tau_{0:t}$ as described in the paper but in Eq. (4) $q_\phi$ also includes  dependence on $\theta_{t-1}$. This dependence is not clear, does this represent the time-evolving hidden state of the LSTM, $h_{t-1}$?

1. The one-step prediction encoders in Eq. (6) and (7) should have dependence on $\theta^s_{t+1}$ and $\theta^r_{t+1}$, respectively, as specified in the FN-MDP. Though Fig 2(d) shows that state and reward prediction help improve performance, predicting  $(s_{t+1},r_{t+1})$ from $(\theta^s_{t}, \theta^r_{t})$ seems to ignore the latent change factor dynamics. A clarification from the authors would be helpful.

1. The multiple loss functions terms each have an associated weight, which seems like a hyper-parameter tuning nightmare. This can hurt the applicability of the proposed model in practice. Could the author comment on this aspect? On a related note, Section D.4.1 specifies the values used for these weights but is missing a description of the hyper-parameter selection method.

1. The experiments verifying the values of learned reward CFs are insightful. It would be interesting to verify the dynamics CF in a similar experiment.

**Minor comments:**

- Eq. (2) seems to have a typing error, $r_t$ should be function of $s_t,a_t$ instead of $s_{t-1},a_{t-1}$.
- Line 87 typing error, $c$ should not have subscript $i$.
- Line 93-94 is confusing and seems to be making contradictory claims.
- Algorithm 1, initial values of $\theta_{old}$ not specified.
- Algorithm 1, line 3, what is $k$? What does $\tau^i_{0:k}$ denote?
- The smoothing technique applied is slightly unusual. Based on my reading, most works employ moving average-based smoothing.
- Figure 2(f) has incorrectly labelled axes.
- Appendix lines 190-191 state that CF inference networks are fully-connected, shouldn’t they be LSTMs?
- There is a lot of notation used in the paper, which naturally introduces scope of typing errors. I assume such issues will be fixed.

**Limitations:**

The paper provides a brief discussion on some limitations, namely the scalability and applicability of the approach to complex problems. The appendix provides a more detailed discussion on the effect of different assumptions or inductive biases, and acknowledges the lack of theoretical guarantees for the proposed method.

**Strengths And Weaknesses:**

Learning effective and stable policies in non-stationary environments is an active area of research within the community. This is an important area of research and contributions in this direction can facilitate the application of deep RL to real-world scenarios. This work formalizes a non-stationary analogue of factored MDPs and proposes algorithms to model the environment and combine it with policy optimization to provide a general-purpose framework. The presented approach seems reasonable and theoretically sound. To the best of my knowledge, this represents novel and original work.

The writing is clear and concise. The factored generative model is quite complex with many moving parts, but the explanation and the diagrams make things fairly straightforward to follow. It is appreciable that both continuous and discrete changes are considered, and the framework is also adapted to image observations. The experiments are suitable and include comparison with recent works. The extensive results both in the main paper and the appendices are impressive. Notably, the authors perform significance tests for comparison with previous work.

The paper presents good quality and original work, with clear explanation of concepts and extensive results. The quality of submission can be enhanced by addressing some clarification questions and suggestions, provided below.

---

> ### Author Response · Authors · 2022-07-30
> **Response to Reviewer rs6w (1/2)**
>
> Thanks for the careful review, we have made some changes in the revised version (highlighted in blue). We answer the points one by one in the following:
>
> > Q1: The inference networks infer $\theta_{t}$ from $\tau_{0: t}$ as described in the paper but in Eq. (4) $q_{\phi}$ also includes dependence on $\theta_{t-1}$. This dependence is not clear, does this represent the time-evolving hidden state of the LSTM, $h\_{t-1}$ ?
>
> Thanks for pointing this out, this was a typo on our side. Yes, $\mathbf{\theta}\_{t-1}$ in $q_{\phi}$ represents the hidden state of the LSTM $\mathbf{h}\_{t-1}$. We have changed it into  $q_{\phi}(\mathbf{\theta}_{t}\mid\mathbf{s}_t, \mathbf{a}_t, r_t, \mathbf{h}\_{t-1})$ in the revised version (updated Eq. (4)).
>
> > Q2: The one-step prediction encoders in Eq. (6) and (7) should have dependence on $\theta\_{t+1}^{s}$ and $\theta_{t+1}^{r}$, respectively, as specified in the FN-MDP. Though Fig 2(d) shows that state and reward prediction help improve performance, predicting $\left(s\_{t+1}, r\_{t+1}\right)$ from $\left(\theta_{t}^{s}, \theta_{t}^{r}\right)$ seems to ignore the latent change factor dynamics. A clarification from the authors would be helpful.
>
> We clarify in the revised version that we only use the one-step prediction loss, when we expect the changes to be smooth.
> - In the discrete changes case, we do not use the prediction losses at the timesteps  $(\tilde{t}_1-1, \ldots, \tilde{t}_M-1)$ when there are discrete changes happening  at timesteps $\boldsymbol{\tilde{t}} = (\tilde{t}_1, \ldots, \tilde{t}_M)$, since the changes are not smooth by definition.
> - For the continuous changes, we have two settings: across-episode and  within-episode. In the across-episode changes, we do not use the one-step prediction for the first time-step in the next episode, because the state is randomly initiated in each episode. In the within-episode changes, we use the prediction loss, but we ignore the latent change factor dynamics, because we assume that the changes are smooth across time.
>
> We tested empirically if adding the latent change factors $\left(\mathbf{\theta}\_{t+1}^{s}, \mathbf{\theta}\_{t+1}^{r}\right)$ would help in the prediction of future states and rewards. In particular, we tried this ablation in a setting with continuous changes on the dynamics in Minitaur. The results below show that using either $\theta^{t+1}$ or $\theta^{t}$ seems to have similar performances on rewards. We will include this ablation on all continuous within-episode settings into our final version.
>
> |        Methods       | Average final rewards |
> |:--------------------:|:----------------------:|
> | Using $\mathbf{\theta}^{t+1}$ |      5.9 (+/-11.7)     |
> |  Using $\mathbf{\theta}^{t}$  |      6.3 (+/-10.4)     |
>
>
> > Q3: The multiple loss functions terms each have an associated weight, which seems like a hyper-parameter tuning nightmare. This can hurt the applicability of the proposed model in practice. Could the author comment on this aspect? On a related note, Section D.4.1 specifies the values used for these weights but is missing a description of the hyper-parameter selection method.
>
> Thanks for pointing this out. We have clarified in the revised version that we use the automatic weighting method in [Liebel et al 2018] to learn the weights for $k_1, \ldots, k_5$ and grid search for $w_1, \ldots, w_7$. We have also added more details in the Supplementary Section D.4.1. We believe these are common hyper-parameters selection strategies, so they do not limit the applicability of the proposed model.
>
> [Liebel et al 2018] Liebel, Lukas, and Marco Körner. "Auxiliary tasks in multi-task learning." arXiv preprint arXiv:1805.06334 (2018).
>
> > Q4: The experiments verifying the values of learned reward CFs are insightful. It would be interesting to verify the dynamics CF in a similar experiment.
>
> Thanks for the suggestion. We verify one dynamic CF (wind forces in Half-Cheetah). We randomly sample a few data points and compute the Euclidean distance below. $D(f^{i,j}_w)$ and $D(\theta^{s}\_{i,j})$ denotes the Euclidean distance on wind force $f_w$ and $\theta^s$ between two sampled data points. We can find that there is a positive correlation between the distance of learned $\theta^s$ and true wind forces. We added two heatmaps (similar to Fig. 2(f)) in the revised Fig. A10 in the Appendix.
>
> | $D(f^{i,j}_w)$ | $D(\theta^{s}_{i,j})$ |
> |:--------------:|:---------------------:|
> |      1.62      |          2.9          |
> |      10.12     |          6.8          |
> |      11.74     |          8.5          |
> |      14.42     |          11.0         |
>
> > Q5: Minor points on notations
>
> Thanks for the point out the typos in the text and figures. We have corrected them in the revised version.

---

> > ### Author Response · Authors · 2022-07-30
> > **Response to Reviewer rs6w (2/2)**
> >
> > > Q6: Line 93-94 is confusing and seems to be making contradictory claims.
> >
> > We have added the following explanation in the revised version.
> >
> > "Although $\mathbf{c}^{\cdot \rightarrow \cdot}$ and $\epsilon$ are stationary, we model the changes in the functions and some changes in the graph structure through $\mathbf{\theta}$.
> > For example a certain value of $\mathbf{\theta}_t^r$ can switch off the contribution of some of the state or action dimensions in the reward function, or in other words nullify the effect of some edges in $\mathcal{G}$. Similarly the contribution of the noise distribution to each function can be modulated via the change factors. On the other hand, this setup does not allow adding edges that are not captured by the binary masks $\mathbf{c}^{\cdot \rightarrow \cdot}$."
> > Hopefully this clarifies things a bit. Given the space constraints, this explanation might be a bit short, so we will expand on this concept in the final version with an additional page.
> >
> > > Q7: Algorithm 1, initial values of $\theta_{\text {old }}$ not specified. line 3, what is $k$ ? What does $\tau_{0: k}^{i}$ denote?
> >
> > We have defined $\mathbf{\theta}\_{\text {old }}$ and  $k$ in updated Algorithm 1. $\tau\_{0: k}^{i}$ denotes the $i$-th collected trajectory with length $k$ for estimating the FN-VAE (See Algorithm A1).
> >
> > > Q8: The smoothing technique applied is slightly unusual. Based on my reading, most works employ moving average-based smoothing.
> >
> > We have added an experiment to compare with moving average-based smoothing. Specifically, we choose the moving average and exponential moving average smoothing with the different hyper-parameters:
> >
> > - Moving average: $\mathcal{L}\_{\text {smooth }}=\sum\_{t=2}^{T}\left(\left\|\|\mathbf{\theta}_{t}-(\mathbf{\theta}\_{t-1}+\mathbf{\theta}\_{t-2}+\ldots+\mathbf{\theta}\_{t-T})/T\right\|\|\_{1}\right)$;
> >
> >
> > - Exponential Moving average: $\mathcal{L}\_{\text {smooth}}=\sum_{t=2}^{T}\left(\left\|\|\mathbf{\theta}_{t}-(\beta \mathbf{\theta}\_{t-1}+(1-\beta) \mathbf{v}\_{t-2})\right\|\|\_{1}\right)$, where $\mathbf{v}\_{t}=\beta \mathbf{\theta}\_{t} + (1-\beta)\mathbf{v}\_{t-1}$ and $\mathbf{v}\_{0}$ is a zero vector.
> >
> > We also report the results of the experiments on Half-Cheetah with continuous changes on dynamics. The results are given below.
> >
> >
> > |                  Methods                 | Average final rewards |
> > |:----------------------------------------:|:------------------------:|
> > |      Moving average smoothing $(T=2)$      |      -23.7 (+/-19.6)     |
> > |      Moving average smoothing $(T=4)$      |      -25.9 (+/-20.4)     |
> > |      Moving average smoothing $(T=8)$      |      -25.6 (+/-17.5)     |
> > |  Exponential moving average $(\beta=0.9)$ |      -31.5 (+/-25.6)     |
> > | Exponential moving average $(\beta=0.98)$ |      -26.2 (+/-20.3)     |
> > |                   Ours                   |      -24.8 (+/-21.1)     |
> >
> > The results suggest that our smoothing term has the similar performances with the moving average smoothing.
> >
> > > Q9: Appendix lines 190-191 state that CF inference networks are fully-connected, shouldn't they be LSTMs?
> >
> > Sorry for the ambiguity. Here we mean the LSTM layers are followed by those dense layers. We have clarified this in updated Appendix.

---

> > > ### Comment · Reviewer_rs6w · 2022-08-08
> > > **Thank you for the response**
> > >
> > > Thank you for the detailed response, which addresses all of my concerns. After reading the other reviews, related discussions, and re-reading the paper, I have increased my score.

---

> > > > ### Author Response · Authors · 2022-08-09
> > > > **Thank you for the feedback**
> > > >
> > > > We highly appreciate your positive and valuable feedback. As NeurIPS allows an additional content page for the camera-ready version, we will include the results of using different smoothness losses (given in the rebuttal) in the evaluation section of our final version if accepted.

---

### Meta-Review · Area_Chair_fJSf · 2022-08-26

**Recommendation:** Accept
**Confidence:** Certain

**Metareview:**

The paper proposes a factored reinforcement-learning method to deal with non-stationary environments.
After reading the authors' rebuttals, the reviewers agree that this paper provides an original and sound contribution that deserves publication.
We recommend that the authors modify their paper as reported in their answers to the reviewers' comments.

**Award:**

No

---

### Decision · Program_Chairs · 2022-09-14

Accept